# LaDiR: Latent Diffusion Enhances LLMs for Text Reasoning

**Haoqiang Kang**[1]    **Yizhe Zhang**[2]    **Nikki Lijing Kuang**[1]    **Nicklas Majamaki**[1]

**Navdeep Jaitly**[2]    **Yi-An Ma**[1]    **Lianhui Qin**[1]

[1]University of California, San Diego    [2]Apple

## ABSTRACT

Large Language Models (LLMs) demonstrate their reasoning ability through chain-of-thought (CoT) generation. However, LLM's autoregressive decoding may limit the ability to revisit and refine earlier tokens in a holistic manner, which can also lead to inefficient exploration for diverse solutions. In this paper, we propose *LaDiR* (**La**tent **Di**ffusion **R**easoner), a novel reasoning framework that unifies the expressiveness of continuous latent representation with the iterative refinement capabilities of latent diffusion models for an existing LLM. We first construct a structured latent reasoning space using a Variational Autoencoder (VAE) that encodes text reasoning steps into blocks of thought tokens, preserving semantic information and interpretability while offering compact but expressive representations. Subsequently, we utilize a latent diffusion model that learns to denoise a block of latent *thought tokens* with a blockwise bidirectional attention mask, enabling longer horizon and iterative refinement with adaptive test-time compute. This design, combined with explicit diversity guidance during diffusion inference, enables the generation of multiple diverse reasoning trajectories that explore distinct regions of the latent space, rather than producing repetitive solutions as often occurs in standard autoregressive sampling. We conduct evaluations on a suite of mathematical reasoning and planning benchmarks. Empirical results show that LaDiR consistently improves accuracy, diversity, and interpretability over existing autoregressive, diffusion-based, and latent reasoning methods, revealing a new paradigm for text reasoning with latent diffusion.

## 1 INTRODUCTION

Large language models (LLMs) have demonstrated remarkable reasoning abilities through extensive pretraining on human languages, yet the inherent limitations of the autoregressive (AR) paradigm are becoming increasingly difficult to overlook (Zhou et al., 2024b; Bachmann & Nagarajan, 2025). As shown in Fig. 1 (top left), their sequential nature prevents revising earlier tokens, making self-refinement inefficient and difficult (Chen et al., 2024; Huang et al., 2024). Moreover, AR models with discrete CoT generate a linear chain of thought (CoT) (Dziri et al., 2023; Wei et al., 2023), which limits reasoning diversity and restricts exploration of multiple valid solutions (Naik et al., 2024; Yu et al., 2024).

Diffusion models (Ho et al., 2020), originally introduced for generation in continuous domains like images, have recently gained attention in text generation for their ability to maintain global coherence and enable iterative refinement (Ye et al., 2024b; Nie et al., 2025; Lou et al., 2023; Yu et al., 2025b; Weligalle, 2025; Sahoo et al., 2024; Gulrajani & Hashimoto, 2023). Moreover, prior works have explored continuous or latent diffusion for language generation (Li et al., 2022; Lovelace et al., 2024; Zhang et al., 2023; Lovelace et al., 2026; Cetin et al., 2025), operating diffusion in latent spaces obtained from text autoencoders or token-embedding spaces. Existing works largely emphasize the parallelization properties of diffusion models (Israel et al., 2025; Nie et al., 2025; Weligalle, 2025) or evaluate fluency in text generation (Li et al., 2022; Lovelace et al., 2024; Zhang et al., 2023; Lovelace et al., 2026). Arguably, a more important direction is to ask: *How can these approaches enhance the reasoning capabilities of LLMs?* We focus on one particularly promising capability: *the ability to self-correct and refine reasoning chains at semantic levels in latent space*. As shown

in Fig. 1 (top right), such self-refinement cannot be achieved by discrete diffusion language models that merely transit into masked tokens.

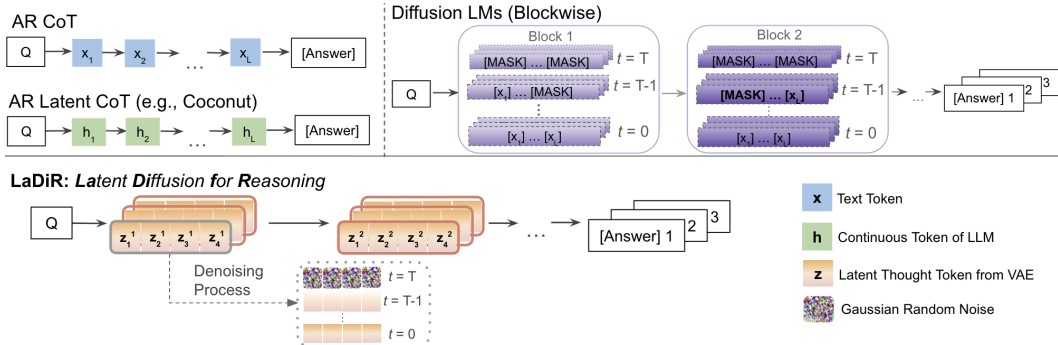

Figure 1: Comparison of reasoning paradigms: autoregressive CoT and latent CoT generate discrete or continuous tokens sequentially; diffusion LMs iteratively convert [MASK] tokens to discrete text tokens in parallel in a semi-autoregressive way; and our proposed method LaDiR reasons over *latent* thought tokens via diffusion, enabling iterative refinement at semantic level and diverse exploration.

To address this limitation, we introduce LaDiR (**La**tent **Di**ffusion **R**easoner), a flexible reasoning framework that encodes high-level semantic representations of reasoning steps into continuous latent tokens via a Variational Autoencoder (VAE) as *latent thought tokens*, and trains a latent diffusion model over them to perform reasoning. This bridges the gap between surface-level token refinement and deeper semantic reasoning. After the reasoning process, the model generates final answer tokens conditioned on the generated latent thought tokens. Unlike prior latent diffusion works for text generation (Li et al., 2022; Lovelace et al., 2024; Zhang et al., 2023), which focus on fluent text generation, our framework is explicitly designed for latent *reasoning*: it learns causal dependencies across reasoning steps through blockwise diffusion, propagates answer correctness signals back to latent tokens.

Our proposed paradigm establishes a new reasoning framework as a post-training method, bringing several distinctive advantages. First, the iterative refinement ability of diffusion enables a better trade-off between accuracy and test-time compute, as additional denoising steps can be flexibly allocated to improve performance. Second, our framework introduces a diversity-guidance mechanism that applies repulsive forces during diffusion inference, pushing latent trajectories apart within a batch to explore multiple *diverse* reasoning paths, where as AR models tend to collapse to similar trajectories. Finally, leveraging a VAE-based latent space enhances interpretability over continuous diffusion models, making the reasoning process more transparent and readable.

Experimentally, we demonstrate that diffusion-based latent reasoning is not only more accurate but also qualitatively different from prior approaches. On math reasoning benchmarks, including GSM8K (Cobbe et al., 2021) and MATH (Hendrycks et al., 2021), where Coconut (Hao et al., 2024) fails to surpass AR CoT supervised finetuning (SFT), LaDiR consistently outperforms it on average across 7 benchmarks with the LLaMA 3.1 8B model (Dubey et al., 2024). This suggests that modeling reasoning at the *semantic level*, rather than at the token level, may lead to more faithful intermediate steps that accumulate into stronger final answers. Moreover, on the Countdown planning task, LaDiR shows over 30% absolute improvement in both Pass@1 and Pass@100, indicating that latent thought tokens potentially enhance global planning ability, while *parallel diversity exploration* enables the model to generate diverse reasoning paths. Together, these findings suggest that diffusion-based latent reasoning provides a principled way to balance accuracy and diversity—key ingredients for advancing beyond sequential autoregressive reasoning.

## 2 PRELIMINARIES

This section introduces key concepts and notations in VAE and latent diffusion models (Rombach et al., 2022). Detailed formulations and background information about VAE and diffusion models are provided in Appendix B.

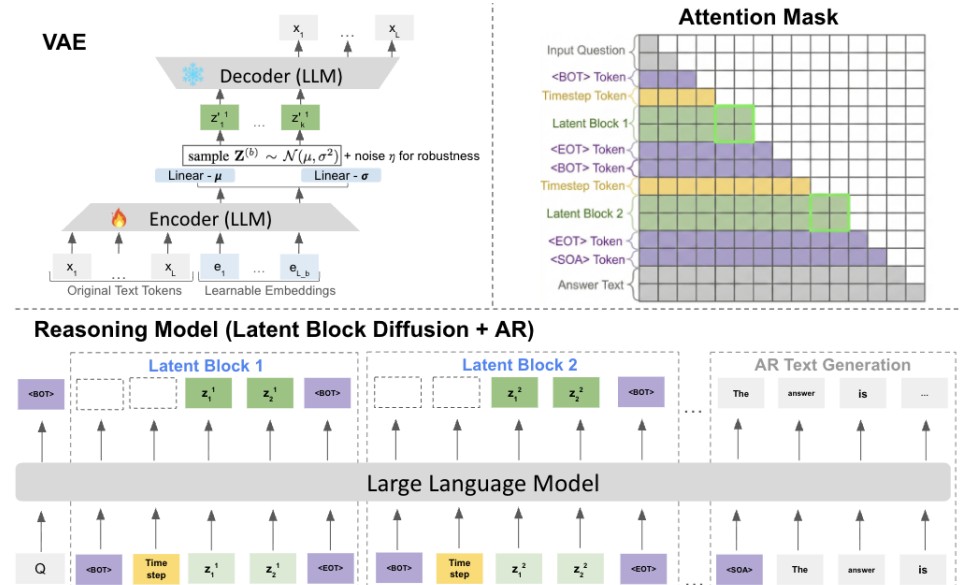

Figure 2: Illustration of our block-wise latent reasoning framework. A question $Q$ is first input as condition to generate latent blocks, each delimited by <BOT> and . For each block, the model iteratively denoises latent tokens $\hat{\mathbf{Z}}^{(b)}$ across timesteps, with bidirectional attention inside a block and causal attention across blocks. The reasoning process terminates when the model emits the <SOA> token, after which the model generates the answer text autoregressively.

## 2.1 VARIATIONAL AUTOENCODER

A Variational Autoencoder (VAE) (Kingma & Welling, 2013) learns a latent representation of data by balancing reconstruction accuracy and prior regularization. Let $x \in \mathbb{R}^{L \times d_x}$ denote a sequence of token embeddings with length $L$ and embedding dimension $d_x$, and $z \in \mathbb{R}^{M \times d_z}$ denote latent representations with $M$ latent tokens of dimension $d_z$. We adopt the $\beta$-VAE (Higgins et al., 2017), where a scaling factor $\beta$ controls this trade-off: $\mathcal{L}_{\beta\text{-VAE}} = \mathbb{E}_{q_\phi(z|x)}[-\log p_\theta(x|z)] + \beta \operatorname{KL}(q_\phi(z|x) \| p(z))$

Here, $q_\phi(z|x)$ is the encoder distribution parameterized by $\phi$, $p_\theta(x|z)$ is the decoder likelihood parameterized by $\theta$, and $p(z) = \mathcal{N}(0, I)$ is the prior distribution over latents.

Larger $\beta$ values encourage disentangled and structured latent spaces, at the cost of reconstruction fidelity. During inference, the encoder of VAE produces a mean/variance pair $\{(\mu, \sigma)\}$, and a latent token $z$ is sampled as $z = \mu + \sigma \odot \epsilon, \ \epsilon \sim \mathcal{N}(0, I)$.

## 2.2 LATENT DIFFUSION AND FLOW MATCHING

Latent diffusion models (Ho et al., 2020; Rombach et al., 2022) generate data by denoising latent variables from Gaussian noise in the latent space of a VAE, which preserves high-level semantic structure. Diffusion can also be viewed as a continuous-time generative flow trained via *flow matching* (Lipman et al., 2022), which we adopt as our primary framework for its superior performance. The training and inference processes are as follows:

**Training** Let $\{z_t\}_{t \in [0,1]}$ denote a path interpolating between clean data $z_0 \sim p_{\text{data}}$ and noise $\epsilon \sim \mathcal{N}(0, I) : z_t = (1 - t)z_0 + t\epsilon$. This path is controlled by an ordinary differential equation (ODE) $u^\star(z_t, t) = \frac{dz_t}{dt} = \epsilon - z_0$, where $u^\star$ is the target velocity field. A neural network $u_\theta(z_t, t)$ is trained to approximate $u^\star$ by minimizing the flow matching loss:

$$\mathcal{L}_{\text{FM}} = \mathbb{E}_{t \sim \mathcal{U}(0,1), \, z_0 \sim p_{\text{data}}, \, z_1 \sim \mathcal{N}(0,I)}\left[\|u_\theta(z_t, t) - u^\star(z_t, t)\|^2\right]. \tag{1}$$

**Inference.** At generation time, the process begins from Gaussian noise $z_1 \sim \mathcal{N}(0, I)$. The learned velocity field $u_\theta(z_t, t)$ is then integrated backward in time using an ODE solver as follows: $z_{t-\Delta t} = z_t - \Delta t \, u_\theta(z_t, t)$, with steps from $t = 1$ to $t = 0$. The final state $z_0$ corresponds to a clean latent representation. This procedure naturally supports *iterative refinement*, as each integration step progressively transforms noise into a coherent latent $z$.

## 3 METHODOLOGY

Our approach separates reasoning from answering. A variational autoencoder (VAE) constructs a latent space of intermediate reasoning steps, encoding each step as a block of *thought tokens*. We further utilize a reasoning model that predicts and refines *thought tokens* via latent diffusion, and then generates the final answer tokens conditioned on the denoised latent tokens.

### 3.1 ARCHITECTURE

We employ a VAE to construct the latent space of intermediate reasoning steps, and a reasoning model that predicts latent tokens via diffusion and generates the final text answer.

**Blockization.** We separate the chain-of-thought (CoT) reasoning and the final answer in the dataset using the prefix ''The answer is''. The text preceding the prefix is treated as CoT $c$, while the text following is treated as the final answer $y$. We then split $c$ into individual sentences, each treated as a *block* of latent tokens with block size $L_b$: $\mathbf{Z}^{(b)} = \{z_1^{(b)}, \ldots, z_{L_b}^{(b)}\}, \quad b = 1, \ldots, N$. This one-sentence-per-block design ensures that each reasoning step is localized in latent space, diffusion is performed at the block level as block diffusion Arriola et al. (2025).

**VAE architecture.** As shown in Figure 2 (left), our VAE encoder is initialized from a pretrained LLM and fine-tuned with all parameters, along with $L_b$ learnable embeddings. The encoder's last hidden state is passed through two linear projections to obtain the mean $\mu$ and variance $\sigma^2$, from which we sample $\mathbf{Z}^{(b)} \sim \mathcal{N}(\mu, \sigma^2)$. The decoder is a *frozen* pretrained LLM that conditions on the sampled $\mathbf{Z}^{(b)}$ to reconstruct the corresponding block of text. This design enables the encoder to compress each reasoning step into a structured latent representation aligned with the semantic space of the language model.

**Reasoning model architecture.** We utilize an existing LLM as our reasoning model. As illustrated in Fig. 2, consider the prediction of the second latent block. After the input question $Q$, we insert a special token <BOT> to mark the start of a block, followed by the first block tokens $\mathbf{Z}^{(1)}$, and a token  to mark its end. For the second block, since it is being predicted, we add a timestep embedding between <BOT> and $z_1^{(2)}$ to encode the timestep information. Once the latent reasoning process is complete, we switch to text generation mode by appending a <SOA> token to indicate the start of the answer, which is then generated autoregressively. To balance lookahead and variable-length generation, we adopt a hybrid attention mask $\mathcal{M}$ (Fig. 2, top right). Within each block, tokens attend *bidirectionally*, enabling the model to internally *reason* over a horizon defined by the block size and capture richer local dependencies. Across blocks, attention is strictly *causal*, so later steps depend on earlier ones in an autoregressive manner.

### 3.2 TRAINING

We train the two components separately: the VAE is first trained to learn latent representations of *thought tokens*, after which the reasoning model is trained to predict these *thought tokens*. We describe each stage in turn, beginning with VAE training and followed by reasoning model training.

#### 3.2.1 VAE TRAINING

We build on the standard $\beta$-VAE training and inference framework described in Section 2.1, with the following adaptations tailored to our task.

**Robustness augmentations.** To improve generalization and make the latent space resilient to noise and input variability, we introduce two augmentation strategies during training:

- **Latent Gaussian noise.** For each latent token $z_i^{(b)}$, we inject isotropic Gaussian perturbations: $z_i'^{(b)} = z_i^{(b)} + \eta_i, \quad \eta_i \sim \mathcal{N}(0, k^2 I)$, where we find $k = 3$ achieve the best downstream performance. This enhances robustness by smoothing the latent space and mitigating sensitivity to small semantic variations.

- **Input token substitution.** For the encoder input sequence, with probability $p = 0.3$ we replace a token with another randomly chosen token (sampled uniformly from the LLM vocabulary). This forces the encoder to learn invariances to paraphrasing, typos, or minor corruptions in input text, ensuring that latent representations capture semantic content rather than exact lexical form.

Together, these augmentations encourage the VAE to build a smoother latent space that is both robust to perturbations and expressive enough to encode thought-level reasoning steps. A more detailed diagram of the VAE can be seen in Appendix E.

### 3.3 REASONING MODEL TRAINING

After constructing a latent reasoning space with the VAE, we train a latent diffusion model $f_\psi$ from the same pretrained LLM as in our VAE model to denoise latent blocks, gradually transforming noisy latent representations to coherent reasoning blocks. Empirically, we observe that training with the flow-matching objective yields the best performance (see Appendix B.4), and therefore adopt it as our default training objective in the paper.

**Answer Token Loss.** While $f_\psi$ learns to predict latent reasoning trajectories, to avoid explicitly decoding these steps through the VAE decoder for efficiency during inference, we use the same transformer backbone $\psi$ with a LM head to autoregressively predict answer text tokens conditioned on the latent reasoning blocks. To this end, given the question $q$, the reasoning blocks $\mathbf{Z}^{(\leq B)}$, and the past answer tokens $y_{<w}$, the model predicts the next answer token $y_w$ with distribution $p_\psi(y_w \mid q, \mathbf{Z}^{(\leq B)}, y_{<w})$. The training objective for those answer tokens is the cross-entropy loss:

$$\mathcal{L}_{\text{Ans}} = -\sum_{w=1}^{W} \log p_\psi(y_t \mid q, \mathbf{Z}^{(\leq B)}, y_{<w}). \tag{2}$$

**Special Token Loss.** To explicitly control the number of latent blocks, we introduce a special binary classification head on top of the same LLM transformer backbone $\psi$. It predicts whether the next block begins with a `<SOA>` (start-of-answer) or `<BOT>` (begin-of-thought) token whenever an `` (end-of-thought) token is generated. Formally, let $\tau$ index positions of `` tokens in the output. For each $\tau$, the model produces a distribution $p_\psi(s_\tau \mid q, \mathbf{Z}^{(\leq B)}, y_{\leq \tau}), s_\tau \in \{\texttt{<SOA>}, \texttt{<BOT>}\}$, and we minimize the corresponding classification loss, which supervises the model to predict special tokens given the question $q$ and latent reasoning blocks up to position $\tau$:

$$\mathcal{L}_{\text{Spec}} = -\sum_{\tau \in \mathcal{T}_{\text{EOT}}} \log p_\psi(s_\tau \mid q, \mathbf{Z}^{(\leq B)}). \tag{3}$$

#### 3.3.1 STAGE 1: TEACHER-FORCING TRAINING

In the first stage, the model is trained under a *teacher-forcing* regime, where it has access to oracle latent blocks produced by the VAE encoder, denoted as $\mathbf{Z}^{(1:B)}$. At every step, these oracle latents are concatenated between special tokens `<BOT>` and `` and provided as context to the flow-matching model $f_\psi$. The overall training objective jointly optimizes flow matching on latent blocks and cross-entropy supervision on both final answers and special tokens:

$$\mathcal{L} = \lambda_{\text{FM}} \mathcal{L}_{\text{FM}} + \lambda_{\text{Ans}} \mathcal{L}_{\text{Ans}} + \lambda_{\text{Spec}} \mathcal{L}_{\text{Spec}}, \tag{4}$$

where $\mathcal{L}_{\text{FM}}$ is defined in Eq. 1.

#### 3.3.2 STAGE 2: ROLLOUT TRAINING

After Stage 1, there is a mismatch between training and inference. During inference, the model must be conditioned on previous self-generated latents without access to oracle latents, suffering from error accumulation issue. To address this issue, Stage 2 adopts an *rollout* training. We keep the same number of blocks $B$ as in the ground truth, but instead of conditioning on oracle latents, the model generates its own latents $\tilde{\mathbf{Z}}^{(1:B)}$ from random noise using a fewer denoising steps (i.e., $50 \rightarrow 10$, following FlowGRPO (Liu et al., 2025)). We keep the gradients on $\tilde{\mathbf{Z}}^{(1:B)}$ during denoising, allowing answer supervision to backpropagate through the trajectory and directly *shape* latent predictions. To avoid latent collapse as in Coconut w/o curriculum learning (Hao et al., 2024), we keep the flow matching loss. Therefore, the training objective is same as Eq. 4.

### 3.4 REASONING MODEL INFERENCE

At inference time, the model generates a chain of latent reasoning blocks and subsequently produces the final answer in text space. The process unfolds in two phases: (i) latent block generation via iterative denoising, and (ii) answer generation via autoregressive decoding.

**Iterative denoising.** Following the inference process of the standard latent diffusion model as in 2.2, for each block, we initialize with a Gaussian noise, and we gradually transforms the noise into a semantically coherent latent reasoning block $\hat{\mathbf{Z}}^{(b)}$.

**Stopping criterion.** Latent block generation continues until the model explicitly predicts the special token `<SOA>` (*start of answer*). This token signals that sufficient reasoning has been performed and the model should transition from block diffusion to final answer generation.

**Answer generation.** Once reasoning terminates, conditioned on the generated latent reasoning sequence $\hat{\mathbf{Z}}^{(1:\hat{B})}$ and the input question $x$, the model predicts output tokens $y = (y_1, \ldots, y_T)$ autoregressively.

**Diversity improvement in parallel.** Unlike AR models that generate a single reasoning trajectory sequentially, our framework can generate multiple diverse reasoning trajectories in parallel within a batch. To encourage exploration of alternative solutions, we incorporate two complementary mechanisms:

1. **Increased initial noise.** By sampling with an increased variance $\tilde{\sigma}^2$ as initial noise at the first denoising step, we broaden the distribution of starting points for latent trajectories.

2. **Diversity gradient guidance.** At each denoising step, we enhance diversity by adding a repulsion term to push the latent tokens in a batch apart. First, we compute a bandwidth parameter $\sigma$ as the median pairwise distance between the latent tokens in a batch at the current step $\sigma = \text{median}_{i<j} \|z_i - z_j\|_2$. The repulsion force field for a latent token $z_i$ is then defined as

$$\mathbf{F}(z_i) = \sum_{j \neq i} 2 \left( 1 - \frac{\|z_i - z_j\|_2^2}{\sigma^2} \right) \exp\left( - \frac{\|z_i - z_j\|_2^2}{\sigma^2} \right) (z_i - z_j), \forall j \leq B. \quad (5)$$

where $z_j$ is any other latent token in the same batch with batch size $B$. We apply strong repulsion at the beginning of inference and gradually decay its effect over time. Specifically, the time-dependent scale is defined as $\gamma_t = \gamma_{\max} \left( \frac{t}{T} \right)$, where $T$ is the total number of inference steps, $t$ decreases from $T$ to 0, $\gamma_{\max}$ is the initial repulsion strength as a hyperparameter.

Finally, the diversity-guided prediction combines the base model output with the repulsion gradient, in a form analogous to classifier-free guidance (Ho & Salimans, 2022): $\hat{z}_{t-1} = f_\psi(\mathbf{x}_t, t, x) + \gamma_t \mathbf{F}(z)$, where $f_\psi(\mathbf{x}_t, t, x)$ is the model's prediction at step $t$.

## 4 EXPERIMENTS

We evaluate LaDiR across three domains: mathematical reasoning (7 benchmarks), code generation (4 benchmarks), and puzzle planning (Countdown), comparing to AR, latent, and diffusion baselines. Our experiments demonstrate its effectiveness on benchmark datasets, while the analyses in Section 4.5 provide further insights of our method. See Appendix D for experimental details.

### 4.1 MATHEMATICAL REASONING

We begin by assessing LaDiR on a range of mathematical reasoning benchmarks, covering both in-domain datasets, where training and test distributions are closely aligned, and out-of-domain benchmarks that require generalization to unseen problems.

**Datasets** We fine-tune pretrained LLMs on the **DART-MATH** dataset (Tong et al., 2024b), a large-scale dataset synthesized to enhance mathematical reasoning. For evaluation, we adopt two in-domain benchmarks, **Math** (Hendrycks et al., 2021) and **GSM8K** (Cobbe et al., 2021), and five out-of-domain benchmarks to assess generalization: **College-Math** (Tang et al., 2024b), **DeepMind-Math** (Saxton et al., 2019), **OlympiaBench-Math** (He et al., 2024), **TheoremQA** (Chen et al., 2023), and **Fresh-Gaokao-Math-2023** (Tang et al., 2024b). Detailed dataset descriptions are provided in Appendix D.

**Baselines** We benchmark LaDiR against a comprehensive set of baselines categorized into three distinct groups. First, for **Masked Diffusion Methods**, we utilize the LLaDA 8B backbone Nie et al. (2025), evaluating both the base model and a version fine-tuned (SFT) on the same CoT dataset. Second, we compare against **Autoregressive Methods** based on LLaMA 3.1 8B Dubey et al. (2024). This category contains traditional supervised baselines—specifically Solution-Only (No CoT) SFT and CoT SFT with different decoding temperature for randomness—as well as recent AR latent reasoning approaches, including Pause Token Goyal et al. (2023), iCoT Deng et al. (2023), Coconut Hao et al. (2024), CODI Shen et al. (2025a), Discrete Latent Su et al. (2025), Soft Token Butt et al. (2025), Soft Think Zhang et al. (2025), and TaH+ Fu et al. (2025). Finally, we compare against prior **Latent Diffusion Methods**, specifically LD4LG Lovelace et al. (2024) and PLANNER Zhang et al. (2023). Detailed baseline descriptions are provided in Appendix D.1.

**Results** As shown in Table 1, LaDiR achieves the strongest overall performance, surpassing recent competitive latent reasoning baselines including CODI, Soft Thinking, and TaH+. On average, it improves Pass@1 accuracy by 1.5% over the best prior approach (TaH+). Crucially, LaDiR

| Method | In-Domain | | Out-of-Domain | | | | | Avg. |
|---|---|---|---|---|---|---|---|---|
| | MATH | GSM8K | Gaokao | DM-Math | College | Olympia | TheoremQA | |
| **Masked Diffusion Methods — LLADA 8B** | | | | | | | | |
| Base Model | 36.2/48.6 | 77.3/82.9 | 10.2/18.5 | 29.8/36.2 | 30.2/49.1 | 3.0/7.7 | 16.6/21.6 | 29.0/37.8 |
| CoT SFT | 39.0/51.0 | 82.3/88.2 | 20.1/36.0 | 43.7/50.7 | 38.9/48.3 | 5.9/10.2 | 20.9/26.3 | 35.8/44.3 |
| **Autoregressive Methods — LLaMA 3.1 8B** | | | | | | | | |
| Sol-Only SFT | 13.3/18.9 | 16.4/20.7 | 0.0/2.8 | 18.2/23.3 | 15.9/20.6 | 4.7/10.9 | 16.9/22.2 | 12.2/- |
| SFT ($\alpha = 0.1$) | 43.8/46.8 | 84.9/86.2 | 31.2/34.7 | 48.4/49.3 | 46.5/51.0 | 10.5/11.1 | 22.0/22.8 | 41.3/42.6 |
| SFT ($\alpha = 0.7$) | 42.9/49.4 | 84.2/88.6 | 30.4/37.5 | 47.5/51.6 | 45.3/54.3 | 9.9/12.6 | 20.9/24.6 | 40.1/45.4 |
| SFT ($\alpha = 1$) | 42.1/51.0 | 83.5/90.1 | 29.7/39.0 | 46.6/53.1 | 44.6/56.0 | 9.4/13.5 | 20.3/26.0 | 39.3/47.1 |
| SFT ($\alpha = 1.2$) | 41.2/52.5 | 82.6/91.8 | 28.9/40.6 | 45.5/54.6 | 43.6/57.8 | 8.9/14.6 | 19.5/27.5 | 38.4/48.8 |
| Pause Token | 27.2/31.4 | 62.4/67.2 | 21.3/25.7 | 26.7/30.8 | 27.2/32.1 | 3.5/7.8 | 8.3/12.9 | 25.2/29.7 |
| iCoT | 35.2/37.9 | 61.8/63.9 | 30.0/32.9 | 30.6/33.0 | 32.6/34.8 | 4.3/7.1 | 11.5/14.0 | 29.4/31.9 |
| Coconut | 37.3/39.3 | 68.3/74.3 | 26.8/29.3 | 33.5/36.9 | 40.2/42.9 | 5.8/6.3 | 11.4/14.9 | 31.9/34.8 |
| CODI | 38.5/45.1 | 76.3/81.7 | 27.5/35.2 | 38.8/44.9 | 43.2/49.0 | 7.6/14.8 | 8.8/15.7 | 34.3/40.9 |
| Discrete Latent | 43.2/47.3 | 83.9/88.6 | 33.3/39.7 | 44.7/49.5 | 47.1/53.7 | 13.3/17.8 | 20.3/28.5 | 40.8/46.4 |
| Soft Token* | 44.7/- | 83.9/- | -/- | -/- | -/- | -/- | -/- | -/- |
| Soft Think | 44.3/46.7 | 83.7/86.6 | 31.2/33.4 | 49.1/51.7 | 45.9/48.0 | 10.4/13.1 | 22.2/24.7 | 41.0/43.5 |
| TaH+ | 46.1/49.4 | 85.9/89.7 | 30.9/34.0 | 48.2/51.8 | 46.7/50.1 | 12.2/15.2 | 24.5/28.4 | 42.0/45.5 |
| **Latent Diffusion Methods — LLaMA 3.1 8B** | | | | | | | | |
| LD4LG | 9.1/15.3 | 32.9/38.4 | 13.2/20.1 | 24.5/29.6 | 17.9/24.3 | 2.8/8.6 | 9.6/15.7 | 15.7/21.7 |
| PLANNER | 5.7/13.1 | 18.7/24.6 | 15.1/21.3 | 19.9/27.6 | 23.6/29.1 | 2.2/9.0 | 10.3/17.3 | 13.6/20.3 |
| **LaDiR** | **46.2/63.7** | **84.8/93.7** | **35.4/45.8** | **52.3/54.2** | **48.6/60.3** | **12.9/15.3** | **24.7/30.7** | **43.5/52.0** |
| –w/o Stage 2 | 30.7/35.8 | 57.8/62.6 | 24.7/30.2 | 32.0/36.9 | 32.8/38.0 | 5.9/10.5 | 11.9/16.9 | 27.9/33.0 |

Table 1: Math reasoning results across in-domain and out-of-domain benchmarks. Each cell reports **Pass@1 (left) / Pass@100 (right)** accuracy. $\alpha$ is the decoding temperature of LLMs. *Results are taken from the original paper; missing Pass@100 results (denoted as "–") indicate that the codebase is not open-sourced for reproduction.

demonstrates better test-time scalability and diversity, achieving the highest Pass@100 across all benchmarks with a 6.1% absolute gain over AR CoT SFT. Compared to text-based CoT and soft-token methods, our latent reasoning yields more robust solutions on complex benchmarks like DM-Math and College, suggesting that our diffusion objective captures abstract reasoning patterns more effectively than previous curriculum-based (e.g., Coconut) or VQ-VAE (e.g., Discrete Latent) approaches. The performance improvement from Stage 2 rollout training further underscores its importance in mitigating error accumulation, contrasting with prior latent diffusion methods like LD4LG and PLANNER which struggle on reasoning tasks. Overall, LaDiR combines the interpretability of CoT with the expressiveness of continuous diffusion, producing generalizable reasoning traces.

## 4.2 CODE GENERATION

We extend our evaluation to the domain of code generation. In this section, we assess the capability of LaDiR to generate executable code by comparing it against a wide range of diffusion-based and autoregressive coding models and training methods.

**Baselines**   We categorize our baselines into four distinct groups to provide a comprehensive evaluation: (1) **Auto-regressive (AR) Coding Models**, including the state-of-the-art Qwen 2.5 Coder 7B Hui et al. (2024) and OpenCoder Huang et al. (2025); (2) **Masked Diffusion Models**, such as LLaDA, Dream, and the specialist Diffu-Coder; and (3) **Looped Latent Reasoning Models**, represented by Ouro 2.6B Zhu et al. (2025b). Additionally, to rigorously isolate the efficacy of our training methodology, we compare against (4) **Reasoning Methods** built upon the same Qwen3-8B-Base Yang et al. (2025) backbone. This control group includes a standard AR SFT baseline, the training-free latent reasoning approach Soft Thinking Zhang et al. (2025), and TaH+ Fu et al. (2025), a strong supervised latent reasoning baseline.

**Datasets**   We construct a SFT dataset by filtering 73k Python samples from the LingCode dataset Codefuse & Team (2025). For evaluation, we employ four widely adopted benchmarks: MBPP Austin et al. (2021), MBPP+ Liu et al. (2023), HumanEval Chen (2021), and HumanEval+ Liu et al. (2023). Please refer to Appendix D.1 for the detailed filtering pipeline and dataset descriptions.

**Results** As detailed in Table 2, LaDiR achieves the highest average performance among all methods. When compared to the reasoning methods sharing the same Qwen3 backbone, LaDiR demonstrates a substantial advantage, yielding an absolute average improvement of 5.2% over the AR SFT baseline. This gain is more obvious on the challenging HumanEval+ benchmark, where LaDiR surpasses SFT by nearly 8%, proving its capability to synthesize robust and correct solutions for complex programming tasks. Furthermore, LaDiR consistently outperforms competing latent reasoning methods, exceeding both the training-free Soft Thinking and the training method TaH+ baselines across all metrics. While specialized models like OpenCoder and Ouro 2.6B show strong performance on MBPP benchmarks, LaDiR establishes perform better on HumanEval and HumanEval+ by more than 10%, which also outperform a strong looped latent reasoning model Ouro 2.6B on these datasets.

| Method | MBPP | MBPP+ | HumanEval | HumanEval+ | Avg. |
|---|---|---|---|---|---|
| **AR Coding Models** | | | | | |
| Qwen 2.5 Coder 7B | 61.6 | 53.0 | 76.9 | 62.9 | 63.6 |
| OpenCoder | 79.9 | **70.4** | 66.5 | 63.4 | 70.1 |
| **Masked Diffusion Models** | | | | | |
| LLaDA | 50.1 | 42.1 | 35.4 | 30.5 | 39.5 |
| Dream | 68.7 | 57.4 | 56.7 | 50.0 | 58.2 |
| Diffu-Coder | 75.1 | 61.9 | 72.0 | 65.2 | 68.6 |
| **Looped Latent Reasoning Models** | | | | | |
| Ouro 2.6B | **80.4** | 66.6 | 78.2 | 70.7 | 74.0 |
| **Reasoning Methods** | | | | | |
| AR SFT | 63.2 | 52.8 | 84.6 | 76.5 | 69.3 |
| Soft Thinking | 64.2 | 53.1 | 85.0 | 75.2 | 69.4 |
| TaH+ | 65.6 | 56.5 | 85.8 | 79.3 | 71.8 |
| **LaDiR** | 66.8 | 59.5 | **87.4** | **84.2** | **74.5** |

Table 2: Code generation performance on MBPP and HumanEval benchmarks (including their Plus versions). We report Pass@1 accuracy (%). The best results for each metric are highlighted in **bold**.

### 4.3 PUZZLE PLANNING – COUNTDOWN

We evaluate the planning ability of our method using *Countdown*, a combinatorial arithmetic game. Given a set of input numbers, the goal is to reach a target in $[10, 100]$ by applying basic operations $\{+, -, \times, \div\}$. Solving a problem thus demands decomposing the target into intermediate subgoals and chaining them correctly. For example, given input numbers $\{97, 38, 3, 17\}$ and target 14, one valid solution is: $97 - 38 = 59$, $59 - 17 = 42$, $42 \div 3 = 14$. Following Gandhi et al. (2024), we construct a dataset of 500k examples, holding out 10% of target numbers for *out-of-distribution* evaluation as test set. We study two settings of growing complexity: CD-4 and CD-5, which use four and five input numbers, respectively.

**Baselines** We compare against: (1) **LLaMA 8B SFT**, an autoregressive baseline finetuned on the same dataset; (2) **LLaDA 8B SFT** (Nie et al., 2025) and (3) **Dream 7B Base** (Ye et al., 2025b) (evaluated without finetuning) as general-purpose diffusion models; and (4) **MGDM** (Ye et al., 2024a), a task-specific diffusion model for Countdown.

**Results** On the Countdown tasks, as shown in Table 3, our method outperforms autoregressive baselines and remains competitive with specialized diffusion models. In CD-4, it improves Pass@1 by more than 25% over LLaMA 8B SFT and over 20% over LLaDA SFT, demonstrating stronger *global planning* ability beyond token-by-token generation, while also delivering the best Pass@100 and over two% higher *diversity* than any baseline. On the more challenging CD-5 task, our model surpasses AR baselines by nearly 30% in Pass@1 and over 30% in Pass@100. In addition, as shown in Figure 6, our pass@k curve rises steeply with $k$, surpassing MGDM at larger $k$. This high pass@k reflects both diverse trajectory exploration and strong potential for reinforcement learning for post-training (Yue et al., 2025).

| Model | CD-4 P@1 | CD-4 P@100 | CD-4 Div. | CD-5 P@1 | CD-5 P@100 | CD-5 Div. |
|---|---|---|---|---|---|---|
| Dream 7B Base* | 16.0 | 24.7 | 4.1 | 4.2 | 10.3 | 5.6 |
| MGDM† | **91.5** | 95.2 | 3.2 | **46.6** | 70.4 | 4.9 |
| LLaDA 8B SFT | 51.2 | 75.2 | 5.4 | 34.4 | 45.2 | 6.2 |
| LLaMA 8B SFT | 46.7 | 65.3 | 3.0 | 8.9 | 15.4 | 3.5 |
| **LaDiR** | 76.6 | **96.4** | **7.3** | 38.5 | **75.2** | **8.9** |

Table 3: Results on Countdown tasks. We report Pass@1, Pass@100, and Diversity (Div.). Diversity is measured as the number of unique valid solutions discovered among 100 samples. Best results are in **bold**, and second-best are underlined. *Dream 7B Base refers to the open-sourced base model without finetuning on this task. †MGDM is a task-specific small discrete diffusion model rather than a general-purpose language model.

### 4.4 ABLATION STUDY

In this section, we study how inference-time stochasticity and diversity guidance affect solution diversity and accuracy by varying (i) the *initial noise scale*, which controls the variance of Gaussian initialization, and (ii) the *maximum diversity scale* $\gamma_{max}$, which regulates the repulsion strength among latent tokens (see Sec. 3.4). We evaluate both the average number of unique solutions and best-of-100 accuracy. Table 3 shows that increasing noise from 1 to 2 improves both diversity and accuracy, but excessive noise (scale 3) harms convergence despite higher diversity. For diversity guidance, removing repulsion ($\gamma_{max} = 0$) yields the lowest diversity, while moderate values (0.3–0.5) strike the best trade-off. Stronger repulsion ($\gamma_{max} \geq 1.0$) further boosts diversity but causes accuracy to drop, suggesting that over-dispersing latents destabilizes reasoning.

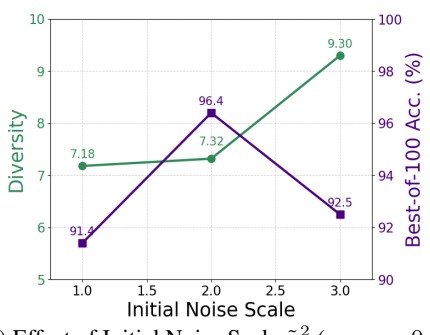
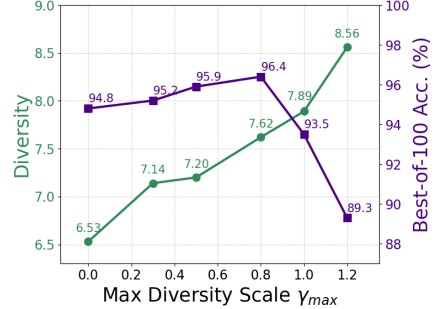

(a) Effect of Initial Noise Scale $\tilde{\sigma}^2$ ($\gamma_{max} = 0.8$).

(b) Effect of Max Diversity Scale $\gamma_{max}$ ($\tilde{\sigma}^2 = 2$).

Figure 3: Ablation study on parameters for diversity during inference on the Countdown-4 dataset.

### 4.5 ANALYSIS

We present analyses on two features of our method: 1) iterative refinement and 2) adaptive test-time compute. Also, we provide additional analyses on inference latency, interpretability and semantic-level reasoning in Appendix C.

**Iterative Refinement** Table 4 shows how the our flow-matching model refines its reasoning across denoising steps. From pure noise at $T = 1$, the model quickly produces structured equations, though early steps contain arithmetic errors (e.g., off-by-one mistakes). As denoising progresses, partial results stabilize—such as $43 + 9 = 52$ appearing consistently from $T = 0$ onward—and later steps are gradually corrected. By $T = 0.25$, the full reasoning matches the ground truth and remains stable through $T = 0$. This demonstrates that our method exhibits the same iterative refinement ability as reasoning models (Shao et al., 2024), progressively correcting previously generated steps. See Table 10 for an example on GSM8K.

| Input | 43, 9, 54, 25, 81 |
|---|---|
| **GT Answer** | 43+9=52, 54-25=29, 52+29=81 |
| Decode($\hat{Z}_{t=1}$) | "I .. ex1 ..." (random noise) |
| Decode($\hat{Z}_{t=0.8}$) | 43+8=51, 54-24=30, 51+30=81 |
| Decode($\hat{Z}_{t=0.7}$) | 43+10=53, 54-25=28, 53+28=81 |
| Decode($\hat{Z}_{t=0.6}$) | 43+9=52, 54-27=27, 52+27=79 |
| Decode($\hat{Z}_{t=0.5}$) | 43+9=52, 54-25=29, 52+28=80 |
| Decode($\hat{Z}_{t=0.4}$) | 43+9=52, 54-25=29, 52+30=82 |
| Decode($\hat{Z}_{t=0.25}$) | 43+9=52, 54-25=29, 52+29=81 |
| Decode($\hat{Z}_{t=0}$) | 43+9=52, 54-25=29, 52+29=81 |

Table 4: Examples of iterative self-refinement of decoded text from the VAE decoder on the Countdown-4 dataset across different denoising timesteps ($t$).

**Adaptive Test-Time Compute.** As shown in Figure 4, using more denoising steps consistently improves accuracy across different math benchmarks. For example, increasing from 5 to 10 steps (a $2\times$ compute increase) yields a large jump of +11.7 points in accuracy on average of 7 benchmarks. Starting from 10 steps, tripling the compute to 30 steps provides an additional +4.8 points on average, while a $5\times$ compute increase to 50 steps brings a total gain of +9.8 points on average. These results demonstrate that our method can flexibly trade test-time compute for higher performance as an alternative paradigm in reasoning for long CoT of existing reasoning models (Jaech et al., 2024; Muennighoff et al., 2025; Liu et al., 2024a; Li et al., 2025). This may motivate adaptive policies that dynamically assign more denoising steps to harder queries, maximizing the overall accuracy–compute trade-off.

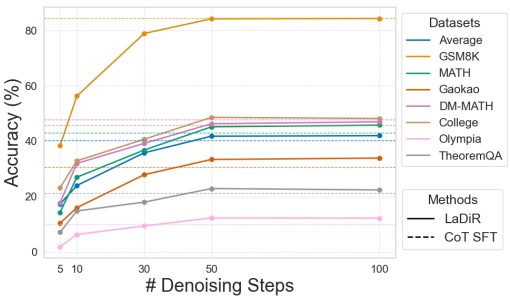

Figure 4: Effect of number of denoising steps on downstream reasoning performance on the math reasoning tasks.

## 5 RELATED WORKS

**Text CoT Reasoning** Chain-of-thought reasoning (Wei et al., 2023) refers to methods which elicit LLMs to generate intermediate reasoning steps in language prior to outputting a final answer in order to improve performance on reasoning tasks. This can be accomplished via prompting methods (Nye et al., 2021; Wei et al., 2023; Khot et al., 2023; Kang et al., 2025; Zhou et al., 2023) or through training LLMs (by SFT, RL, or a combination of the two) to output the intermediate reasoning steps (Yu et al., 2023; Shao et al., 2024; Yu et al., 2025a). Works have also extended CoT to allow LLMs to mimic various tree search algorithms such as BFS or MCTS, which especially improves performance on more complex tasks (Xie et al., 2023; Yao et al., 2023; Zhang et al., 2024; Bi et al., 2025). Beyond following specific algorithms, works that implement long chain-of-thought (combining extensive reasoning, exploration, and reflection) have also demonstrated improved reasoning performance (Shinn et al., 2023; Gandhi et al., 2025; Saha et al., 2025; Xie et al., 2025). One limiting factor with these CoT methods is that they fundamentally work at text token level, constraining the outputs to the token space and limiting the model's horizon.

**Latent Reasoning** Latent reasoning methods address token-level limits of CoTs Wei et al. (2023) by enabling reasoning in a latent space, which only requires higher levels of semantic abstraction (Hao et al., 2024; Pfau et al., 2024; Wang et al., 2024; Zelikman et al., 2024; Jin et al., 2025). Prior works show that reasoning in latent space, rather than discrete tokens, improves performance by allowing LLMs to generate continuous tokens, either self-generated or provided by an auxiliary model (Cheng & Durme, 2024; Liu et al., 2024b; Shen et al., 2025b; Tack et al., 2025; Zhu et al., 2025a; Butt et al., 2025; Zhang et al., 2025; Wu et al., 2025). This has been further extended to recurrent or looped architectures that induce latent reasoning internally, removing the need to represent reasoning steps explicitly as tokens (Chen et al., 2025c; Geiping et al., 2025; Mohtashami et al., 2025; Saunshi et al., 2025; Bae et al., 2025; Zhu et al., 2025b). However, prior latent reasoning approaches lack interpretability, whereas our method constructs the latent space with a VAE, making each step explicit and thus more transparent. Due to the page limit, we discuss further related works in Appendix A.

## 6 CONCLUSION

We introduced LaDiR, a latent diffusion reasoning method that utilizes the iterative refinement capability of latent diffusion models to perform reasoning at the semantic level, our framework offers three key benefits: (1) better tradeoff between accuracy and test-time compute through iterative denoising steps with self-refinement, (2) diverse exploration of reasoning trajectories, and (3) enhanced interpretability through semantically meaningful latent representations. Our experiments on mathematical reasoning, code generation and puzzle planning benchmarks show that LaDiR consistently outperforms AR and diffusion baselines, achieving both better accuracy and diversity in reasoning.

## 7 ACKNOWLEDGMENT

This research is supported by a research grant from Apple. Also, it is partially supported by the NSF award CCF-2112665 (TILOS). It is also supported in part by the CDC-RFA-FT-23-0069 from the CDC's Center for Forecasting and Outbreak Analytics.

## 8 ETHICS STATEMENT

We have read and carefully considered the ICLR Code of Ethics. Our work develops LaDiR , a latent diffusion reasoning method for large language models, with its application scope primarily constrained to mathematical and puzzle-solving domains. We do not believe that this architecture raises any ethical concerns. We affirm that our research and its intended use fully adhere to the ICLR Code of Ethics.

## 9 REPRODUCIBILITY STATEMENT

In order to ensure reproducibility, we release our code for both the VAE training in Section 3.2.1 and the reasoning model (3.3) training in the Supplementary Material. Note that training follows a two-stage pipeline as outlined in Section 3.2: first the VAE is trained and second the latent reasoning model is trained, using the VAE encoder outputs during its training. All datasets used are referenced and publicly available.

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

## A    ADDITIONAL RELATED WORKS

**Diffusion Language Models for Text Reasoning**    Masked diffusion language models attempt to address some common limitations of autoregressive LLMs—such as rigid left-to-right decoding and inefficiency—by iteratively denoising masked tokens, enabling parallel and order-agnostic text generation. Prior studies show that these models achieve better inference efficiency compared to AR models while maintaining comparable performance on both general tasks (Zheng et al., 2024; Gong et al., 2025; Nie et al., 2025; Shi et al., 2025a; Song et al., 2025; Ye et al., 2025c;b) and reasoning benchmarks with chain-of-thought (Gao et al., 2024; Ye et al., 2024b; van Krieken et al., 2025; Ye et al., 2025a). However, these approaches remain constrained to language space, unable to capture reasoning at an abstract semantic level or revise previously generated tokens as continuous diffusion models (Ho et al., 2020; Song et al., 2020) do, and they require training on massive datasets rather than leveraging a well-trained LLM. In contrast, our method overcomes these limitations by structuring reasoning in an interpretable continuous latent space, producing abstract CoT representations with self-correction ability for an existing LLM, while keeping diffusion's strengths in parallel generation to enhance exploration and diversity.

**Latent Diffusion for Language Generation**    Generative text modeling has recently expanded from autoregressive paradigms to diffusion-based approaches that allow for global iterative refinement. One of the first, Diffusion-LM (Li et al., 2022), frames generation as denoising continuous word embeddings to enable fine-grained control, a concept Lovelace et al. (2023; 2026) extended by performing diffusion in a compressed latent space for improved quality and diverse generation modes. For sequence-to-sequence tasks, DiffuSeq (Gong et al., 2022) enables parallel generation with high diversity, while PLANNER (Zhang et al., 2023) addresses long-form text by combining a latent semantic diffusion planner with an autoregressive decoder to reduce repetition. Similarly, Cosmos (Meshchaninov et al., 2025) learns a compressed latent space for diffusion, enabling parallel text generation with robust semantic grounding. In specialized domains, DiffusionDialog (Xiang et al., 2024) utilizes latent variables to handle open-ended conversations, whereas CodeFusion (Singh et al., 2023) and TreeDiff (Zeng et al., 2025) apply diffusion to code synthesis. While prior latent diffusion models focus on text generation, they lack the granularity to model the multistep causal dependencies required for reasoning tasks. We address this by introducing blockwise variable-length diffusion and rollout training, explicitly shifting the objective from generating fluent text to optimizing reasoning trajectories that lead to correct answers.

**Hybrid AR+Diffusion Model Architecture**    Other AR-Diffusion hybrid models have shown successful results in rivaling their AR and diffusion counterparts, particularly in multimodal generation and image understanding. The Transfusion (Zhou et al., 2024a) architecture demonstrated that hybrid models could outperform standard AR models and compete with state-of-the-art diffusion models in image-generation benchmarks, a phenomenon further reinforced by other studies of hybrid models (Fan et al., 2024; Tang et al., 2024a; Xiao et al., 2024). This extends beyond image generation, with several works demonstrating the effectiveness of hybrid AR-diffusion models in other domains such as image understanding, video generation, and robot control (Black et al., 2024; Tong et al., 2024a; Chen et al., 2025a;b). Furthermore–similar to our model architecture–works have demonstrated successful adaptations of frozen models for these hybrid AR-diffusion architectures in multimodal domains (Pan et al., 2025; Shi et al., 2025b). Aside from the difference in domain from these works, many do not use block diffusion for variable-length generations as in LaDiR and we critically introduce CE loss to guide better latent predictions.

## B    ADDITIONAL PRELIMINARIES AND BACKGROUND

We provide more details about the background information of VAE and Diffusion models in this section.

### B.1    VARIATIONAL AUTOENCODER AND $\beta$-VAE

The Variational Autoencoder (VAE) (Kingma & Welling, 2013) is a latent-variable model that learns a compressed representation of data $x$ through an encoder–decoder pair. The encoder $q_\phi(z|x)$ maps inputs into a distribution over latent variables $z$, typically parameterized as a diagonal Gaussian. The decoder $p_\theta(x|z)$ reconstructs the input from $z$, enabling generative sampling. Training maximizes

the evidence lower bound (ELBO):

$$\mathcal{L}_{\text{VAE}} = \mathbb{E}_{q_\phi(z|x)}[-\log p_\theta(x|z)] + \text{KL}\big(q_\phi(z|x) \,\|\, p(z)\big), \tag{6}$$

where the first term ensures faithful reconstruction and the second term regularizes the posterior toward a simple prior $p(z)$, usually $\mathcal{N}(0, I)$. The reparameterization trick,

$$z = \mu_\phi(x) + \sigma_\phi(x) \odot \epsilon, \quad \epsilon \sim \mathcal{N}(0, I),$$

enables low-variance gradient estimates for stochastic optimization.

**$\beta$-VAE.** The $\beta$-VAE (Higgins et al., 2017) introduces a hyperparameter $\beta$ to control the KL weight:

$$\mathcal{L}_{\beta\text{-VAE}} = \mathbb{E}_{q_\phi(z|x)}[-\log p_\theta(x|z)] + \beta \, \text{KL}\big(q_\phi(z|x) \,\|\, p(z)\big). \tag{7}$$

When $\beta > 1$, the model enforces stronger alignment to the prior, which encourages disentangled and interpretable latent variables at the expense of reconstruction fidelity. This property is desirable when latent codes are later used as the substrate for generative modeling.

**Why VAE for Latent Diffusion.** Latent diffusion models (LDMs) (Rombach et al., 2022) operate not on raw high-dimensional inputs (e.g., images or sequences), but in a compressed latent space learned by a VAE. This design provides three key advantages:

1. **Efficiency.** Operating in latent space reduces dimensionality, leading to faster training and inference while maintaining semantic richness.

2. **Semantic abstraction.** The VAE learns to discard imperceptible details and retain high-level structure, making diffusion steps focus on meaningful features rather than pixel-level noise.

3. **Flexibility.** The decoder $p_\theta(x|z)$ ensures that even when denoising occurs in latent space, the final output remains in the original input domain. This separation enables diffusion to generalize across modalities with a shared latent backbone.

## B.2 Latent Diffusion: Training and Inference

Latent diffusion operates in the compressed latent space $z_0$ of a pretrained VAE.

**Forward process.** Noise is added gradually:

$$q(z_t|z_0) = \mathcal{N}\big(z_t; \sqrt{\bar{\alpha}_t}\, z_0, (1 - \bar{\alpha}_t)I\big),$$

with $\bar{\alpha}_t = \prod_{s=1}^{t}(1 - \beta_s)$.

**Training objective.** The denoiser $\epsilon_\theta(z_t, t)$ predicts the injected noise:

$$\mathcal{L}_{\text{LDM}} = \mathbb{E}_{z_0, \epsilon, t}\big[\|\epsilon - \epsilon_\theta(z_t, t)\|^2\big].$$

**Inference.** Generation starts from $z_T \sim \mathcal{N}(0, I)$ and denoises iteratively:

$$z_{t-1} = \frac{1}{\sqrt{\alpha_t}}\Big(z_t - \frac{\beta_t}{\sqrt{1 - \bar{\alpha}_t}}\epsilon_\theta(z_t, t)\Big) + \sigma_t\epsilon.$$

## B.3 Comparison of Parameterizations

Diffusion training can be expressed through different target parameterizations, all of which can be interpreted as variants of the same continuous-time flow. Below we summarize the most common forms:

### B.3.1 $\epsilon$-Prediction

The denoiser directly predicts the added Gaussian noise $\epsilon$:

$$\mathcal{L}_\epsilon = \mathbb{E}_{z_0, \epsilon, t}\Big[\|\epsilon - \epsilon_\theta(z_t, t)\|^2\Big], \tag{8}$$

where $z_t = \sqrt{\bar{\alpha}_t}z_0 + \sqrt{1 - \bar{\alpha}_t}\,\epsilon$. This is the standard DDPM formulation (Ho et al., 2020). It is stable but sometimes less efficient for long horizons.

| Objective | CD-4 Pass@1 (%) |
|---|---|
| MSE Loss | 46.0 |
| $x_0$ | 53.0 |
| $\epsilon$ | 58.0 |
| $v$ | 62.0 |
| $u$ **(ours)** | **73.5** |

Table 5: Ablation study on latent prediction objectives on the Countdown-4 dataset.

### B.3.2  $x_0$-PREDICTION (DDIM-$x_0$)

Instead of noise, the model predicts the clean latent $z_0$:

$$\mathcal{L}_{x_0} = \mathbb{E}_{z_0,t}\Big[\|z_0 - x_{0,\theta}(z_t, t)\|^2\Big]. \tag{9}$$

This corresponds to the DDIM formulation (Song et al., 2020), enabling deterministic sampling and fewer inference steps, but can overfit to data scale.

### B.3.3  $v$-PREDICTION

Proposed by Salimans & Ho (2022), $v$ is defined as a linear combination of noise and clean latent:

$$v = \sqrt{\bar{\alpha}_t}\,\epsilon - \sqrt{1 - \bar{\alpha}_t}\,z_0, \tag{10}$$

with objective

$$\mathcal{L}_v = \mathbb{E}_{z_0,\epsilon,t}\Big[\|v - v_\theta(z_t, t)\|^2\Big]. \tag{11}$$

$v$-prediction is numerically better conditioned, often improving stability across timesteps.

All four parameterizations can be viewed as different instantiations of the same underlying generative flow. $\epsilon$-prediction, $x_0$-prediction, and $v$-prediction specify *which quantity* the denoiser regresses on. Flow matching directly learns the continuous velocity field, avoiding discretization artifacts.

### B.4  BLOCK DIFFUSION

To support flexible and variable-length sequence generation, we employ a *block diffusion* scheme (Arriola et al., 2025) that integrates autoregressive modeling with diffusion. Instead of applying diffusion to individual latent tokens or full sequence, the sequence is divided into contiguous blocks, and diffusion is performed at the block level. This hybrid design retains the open-ended generation of autoregressive models while introducing global coherence within each block.

Suppose we are using $\epsilon$-prediction, and let a sequence be segmented into $M$ blocks $\{B_1, \ldots, B_M\}$, where $B_m \in \mathbb{R}^{k \times d}$ contains $k$ latent tokens of dimension $d$. The forward noising process for block $B_m$ is

$$q(B_{m,t} \mid B_{m,0}) = \mathcal{N}\big(\sqrt{\bar{\alpha}_t}\,B_{m,0}, (1 - \bar{\alpha}_t)I\big), \tag{12}$$

and the denoiser $f_\theta$ is trained to predict the noise at the block level:

$$\mathcal{L}_{\text{block}} = \mathbb{E}_{m,t,\epsilon}\Big[\|\epsilon - f_\theta(B_{m,t}, t)\|^2\Big]. \tag{13}$$

Blocks are generated autoregressively, i.e.,

$$p(B_m \mid B_{<m}) = \int q(B_{m,0} \mid x)$$
$$\times \prod_t p_\theta(B_{m,t-1} \mid B_{m,t}, B_{<m})\, dB_{m,0}. \tag{14}$$

so that each block is denoised iteratively while conditioning on all previously generated blocks.

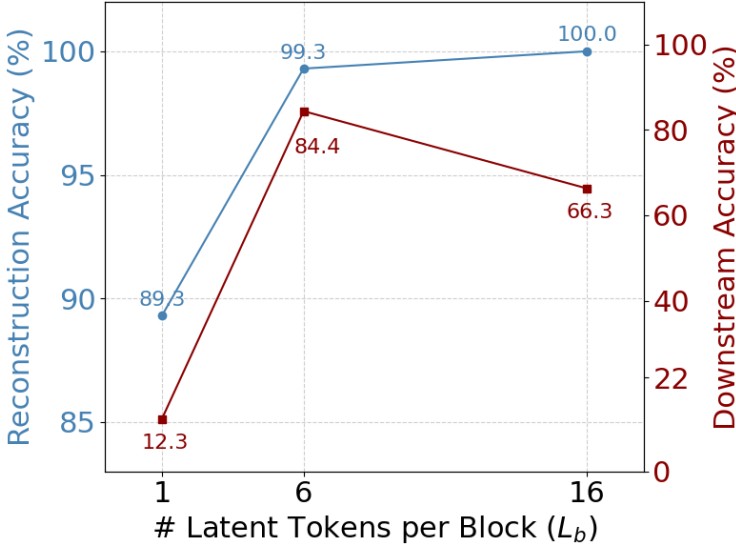

Figure 5: Ablation analysis of block size on the GSM8K benchmark.

**Latent Prediction Objective**  To assess the impact of different training objectives for latent prediction, we compare several widely used formulations. The first baseline is *MSE loss*, which directly minimizes the mean-squared error between predicted and ground-truth latents but yields the weakest results. We then adopt three DDIM-based (Song et al., 2020) objectives: predicting the clean latent state ($x_0$), the added noise vector ($\epsilon$), and the velocity ($v$). These diffusion objectives consistently improve accuracy, highlighting the advantage of explicitly modeling the denoising process rather than relying on direct predictions. Moreover, the flow matching ($u$) objective achieves the strongest gains, suggesting that learning the latent vector field is more effective for capturing the pattern of reasoning compared to DDIM's denoising objectives.

**Effect of the Block Size.**  We investigate how the number of latent tokens per block ($L_b$) influences reconstruction quality and downstream reasoning accuracy on GSM8K. As shown in Figure 5, too few tokens (i.e., 1 token) limit the model's ability to capture necessary information, harming reconstruction. Performance improves as the number of tokens increases, reaching near-perfect reconstruction at $n = 6$. Beyond this point, however, adding more tokens introduces redundancy, which makes the latent diffusion model harder to predict accurately and leads to diminished reasoning accuracy. This reveals a trade-off between compact latent representations and effective downstream reasoning.

**VAE Robustness Augmentations.**  When training our VAE, to improve the robustness of the VAE latent space, we apply two augmentation strategies during VAE training: (1) adding Gaussian noise to latent representations with standard deviation $k$, and (2) randomly substituting input tokens with probability $p$. Table 6 shows the impact of these augmentations on GSM8K accuracy. Best performance is achieved at $k = 3$ and $p = 0.3$. Too little augmentation ($k = 0$ or $p = 0$) results in overfitting to clean inputs, while excessive augmentation ($k = 5$ or $p > 0.5$) degrades the latent space quality.

| Gaussian Noise (p=0.3) | | Token Substitution (k=3) | |
|---|---|---|---|
| $k$ (std) | Acc. (%) | $p$ (prob.) | Acc. (%) |
| 0 | 68.3 | 0.0 | 70.2 |
| 1 | 73.4 | 0.1 | 78.3 |
| **3** | **84.2** | **0.3** | **84.2** |
| 5 | 79.4 | 0.5 | 64.0 |
| – | – | 0.7 | 32.4 |

Table 6: Ablation study on VAE robustness augmentations on GSM8K.

**Blockization Strategy**     We investigate the sensitivity of the model to different blockization strategies by varying the number of sentences per block. Table 7 shows results for 1, 2, and 3 sentences per block. Using more sentences per block requires more latent tokens to maintain reconstruction quality and significantly increases difficulty for the diffusion model. We find that **1 sentence per block** with 4 latent tokens offers the best balance between latent compactness and reasoning accuracy.

| # Sentences | # Latent Tokens | GSM8K | MATH |
|:-----------:|:---------------:|:-----:|:----:|
| **1** | 4 | **84.2** | **45.2** |
| 2 | 8 | 78.4 | 39.6 |
| 3 | 12 | 72.0 | 36.1 |

Table 7: Ablation study on blockization strategy (sentences per block).

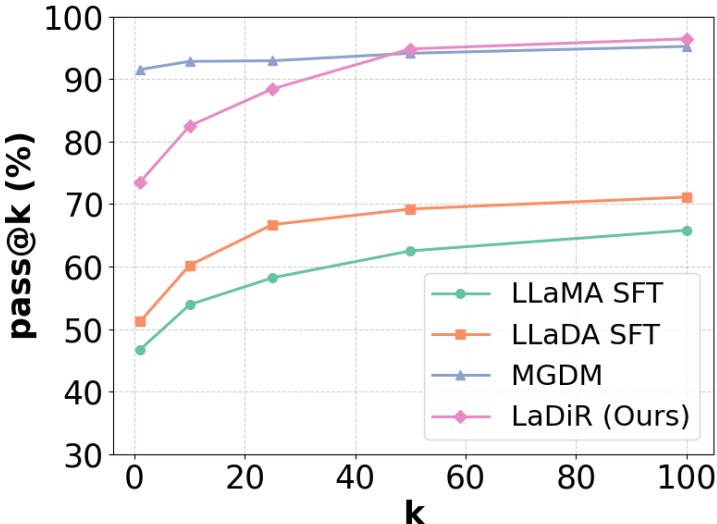

Figure 6: Results for *pass@k* performance on Countdown-4 with $k \in \{1, 10, 25, 50, 100\}$.

## C   ADDITIONAL RESULTS AND ANALYSIS

**Intepretability**    In addition to achieving superior or competitive accuracy across each benchmark, as shown in Table 8, LaDiR also benefits from being more interpretable by nature compared to standard diffusion-based methods.

| Block | Text |
|---|---|
| Question | *Billy sells DVDs. He has 8 customers on Tuesday. The first 3 buy one DVD each, the next 2 buy two DVDs each, the last 3 buy none. How many DVDs did Billy sell?* |
| Decode($Z^{(1)}$): | Billy's first 3 customers buy one DVD each, so that's $3 * 1 = \langle\langle 3 * 1 = 3 \rangle\rangle 3$ DVDs. |
| Decode($Z^{(2)}$): | His next 2 customers buy 2 DVDs each, so that's $2 \times 2 = 4$ DVDs. |
| Decode($Z^{(3)}$): | His last 3 customers don't buy any DVDs, so that's 0 DVDs sold. |
| Decode($Z^{(4)}$): | Therefore, Billy sold a total of $3 + 4 + 0 = 7$ DVDs on Tuesday. |
| Answer | The answer is: 7. |

Table 8: Example of interpretable continuous thought tokens: each latent block $Z^i$ is able to be decoded to human-readable text through the VAE decoder. Each latent block is decoded individually, so the entire latent thought is represented by the block in isolation. This allows for clear interpretibility of each latent thought, while still allowing for a model to reason in a latent space.

**Inference Efficiency**    Table 9 compares inference latency on the MATH dataset using identical hardware and batch size (8). LaDiR with 10 diffusion steps matches the latency of the AR baseline while achieving comparable Pass@1 and higher Pass@100. With 30 steps, LaDiR offers a flexible accuracy-compute trade-off. This efficiency stems from our compact latent representation: each latent block contains only 4 latent tokens, representing on average  22 text tokens, reducing per-step computation and context length compared to autoregressive decoding over long text sequences.

**Reasoning at Semantic Level**    Table 10 demonstrates that LaDiR refines its reasoning through semantic information rather than lexical connections. The pink segments trace how the model in-

| Method | Latency | Pass@1 | Pass@100 |
|---|---|---|---|
| LLaMA CoT SFT | 4.7 | 43.1 | 89.0 |
| **LaDiR (10 steps)** | 4.9 | 42.9 | 90.7 |
| **LaDiR (30 steps)** | 14.8 | 44.9 | 92.8 |

Table 9: Inference latency (seconds/example) comparison on MATH dataset.

crementally refines its intermediate reasoning steps—transforming rough or incomplete arithmetic expressions into logically consistent equations. For example, the model first produces an approximate relation such as $2 = 1 + 1$ and later reconstructs the correct multiplicative form $2 * 1 = 2$, eventually combining these into a complete causal chain that leads to $2 * 4 = 8$. Meanwhile, the green segments serve as stable anchors capturing the essential reasoning cues—phrases like "two times older than" and "four times as old"—which define the causal dependencies between Suzy, Ben, and Brandon. In contrast, the yellow segments correspond to non-reasoning or noisy parts (e.g., "Alice," "Instagram") that the model learns to "ignore" during early denoising, prioritizing semantically coherent and logical grounded reasoning over irrelevant lexical content. In contrast, the SFT baseline remains syntactically fluent but semantically brittle—producing text that appears correct on the surface yet fails to capture the true causal logic. These observations indicate that LaDiR's denoising process operates on the level of semantics and relational consistency, enabling reasoning that aligns with human-like conceptual understanding.

| Question | Brandon's iPhone is four times as old as Ben's iPhone. Ben's iPhone is two times older than Suzy's iPhone. If Suzy's iPhone is 1 year old, how old is Brandon's iPhone? |
|---|---|
| SFT (Baseline) | Suzy's iPhone is 1 year old. Ben's iPhone would be $1 + 1 = 2$ years old. Since Brandon's iPhone is four times as old as Ben's, we compute $4 \times 1 = 4$. Therefore, Brandon's iPhone is 4 years old. The answer is: 4. |

| LaDiR (Ours) | **Block 1** | **Block 2** |
|---|---|---|
| Decode($\hat{Z}_{t=1.0}$) | `10 and 10 10....` | `[garbled tokens]  &^* _` |
| Decode($\hat{Z}_{t=0.9}$) | If Alice is 1 year old, then Bob's age is 2 years older than Alice, which means Bob is 2 = 1 + 1 years old. | ://@natechandra's Instagram is four times as old as Ben's Instagram, so Ben's Instagram is 1 * 4 = 4 years old. |
| Decode($\hat{Z}_{t=0.8}$) | If Suzy's age is 1 year old, then Ben's age is two times older than Suzy's age, which is 2 * 1 = 2 years old. | If Brandon's phone is four times as old as Ben's phone, then Brandon's phone is 2 * 4 = 8 years old.1nbsp;nbsp;nbsp... |
| Decode($\hat{Z}_{t\leq 0.7}$) | If Suzy's iPhone is 1 year old, then Ben's iPhone is two times older than Suzy's iPhone, so Ben's iPhone is 2 * 1 = 2 years old. | If Brandon's iPhone is four times as old as Ben's iPhone, then Brandon's iPhone is 2 * 4 = 8 years old. |
| Answer | The answer is: 8. | |

Table 10: An example of self-refinement during inference on the GSM8k dataset, showing how reasoning becomes progressively clearer as $t$ decreases. Later denoising steps correct arithmetic errors while maintaining earlier structure, demonstrating semantic self-refinement. Pink segments highlight refined reasoning portions, yellow segments indicate non-reasoning or noisy parts that the model gradually corrects, and green segments denote key reasoning cues essential for correct logic.

**Consistency Across Long Reasoning Blocks**  As shown in Table 11, LaDiR maintains coherent and semantically consistent reasoning across a long sequence of blocks. Each block builds logically on the previous one—starting from variable definition ($x$ *as silver coins*), to constraint formulation ($x + (x + 30) = 110$), simplification, and finally the correct numerical conclusion ($x = 40$, *gold* $= 70$). Unlike the SFT baseline, which produces a single-step approximation that conflates intermediate relations, LaDiR preserves arithmetic and causal consistency throughout the reasoning chain, demonstrating stable multi-step inference even with a large number of reasoning blocks.

| Question | Gretchen has 110 coins. There are 30 more gold coins than silver coins. How many gold coins does Gretchen have? |
| --- | --- |
| **SFT (Baseline)** | Let's assume Gretchen has 110 coins in total and 30 more gold than silver. Half of the coins plus 30 should be gold, so $110/2 + 30 = 85$ gold coins. |
| **LaDiR (Ours)** | |
| Decode($Z^{(1)}$) | Let's assume the number of silver coins Gretchen has is $x$ silver coins. |
| Decode($Z^{(2)}$) | We also know that there are 30 more gold coins than silver coins, so the number of silver coins is $x + 30$ gold coins. |
| Decode($Z^{(3)}$) | The total number of coins Gretchen has is $x + (x + 30) = 110$. |
| Decode($Z^{(4)}$) | Combining like terms, we get $2x + 30 = 110$. |
| Decode($Z^{(5)}$) | Subtracting 30 from both sides, we get $2x = 80$. |
| Decode($Z^{(6)}$) | Dividing both sides by 2, we get $x = 40$. |
| Decode($Z^{(7)}$) | Therefore, Gretchen has $30 + 40 = 70$ gold coins. |
| Answer | The answer is: 70. |

Table 11: An qualitative example in GSM8K illustrating the long reasoning blocks generated by our method compared to the baseline SFT.

## D EXPERIMENTAL DETAILS

**Metrics.** We report Pass@1 and Pass@100 accuracy, using an exact string match between the generated arithmetic equations and the ground-truth solution. Pass@k reflects the accuracy that at least one valid solution is found among $k$ samples. In addition, we report Diversity in the Countdown task, measured as the number of unique valid solutions discovered among 100 samples. All models are evaluated with a decoding temperature of 1.0.

### D.1 MATH REASONING

**Implementation Details** The CoT data is segmented into thought-level blocks, where each block corresponds to a single sentence and is represented by 6 latent thought tokens. For VAE training, we set the to $\beta = 10^{-5}$, in which the encoder is finetuned from the backbone model while the decoder remains frozen. The flow-matching model is trained with the objective in Eq. 4, using $\lambda_{\text{FM}} = 5, \lambda_{\text{Ans}} = 1, \lambda_{\text{Spec}} = 2$. During inference, we initialize the Gaussian noise of scale 2, and apply diversity guidance with a maximum scale of 0.8.

**Baselines** We evaluate LaDiR against a comprehensive suite of baselines categorized into three distinct groups:

(1) **Masked Diffusion Models**: We utilize **LLaDA** Nie et al. (2025), a scalable masked diffusion transformer, as the primary baseline. We evaluate both the **Base Model** (zero-shot) and a supervised version (**CoT SFT**) fine-tuned on the same Chain-of-Thought dataset to assess the impact of standard instruction tuning on diffusion architectures.

(2) **Autoregressive (AR) Methods**: Using LLaMA-3.1-8B Dubey et al. (2024) as the backbone, we compare against both standard and advanced latent reasoning approaches.

- **Standard SFT**: We include **Sol-Only SFT** (trained on direct question-answer pairs) and **CoT SFT** (trained with full reasoning traces) to establish lower and upper bounds for standard autoregressive training.

- **Latent Reasoning**: We compare against methods that internalize reasoning: **Pause Token** Goyal et al. (2023) inserts learnable tokens to extend computation; **iCoT** Deng et al. (2023) gradually removes explicit reasoning steps via curriculum learning; and **Coconut** Hao et al. (2024) leverages continuous hidden states as latent thoughts. We also include **CODI** Shen et al. (2025a), which employs self-distillation to compress CoT into continuous space; **Discrete Latent** Su et al. (2025), which utilizes VQ-VAE to encode reasoning into compact discrete codes; **Soft Token** Butt et al. (2025), which utilizes probability-weighted token mixtures; and **Soft Thinking** Zhang et al. (2025), a training-free paradigm operating in a continuous concept space. Finally, we compare with **TaH+** Fu et al. (2025), a strong supervised baseline that dynamically triggers latent iterations only for "hard" tokens using a learned decider.

(3) **Latent Diffusion Models**: To benchmark against prior diffusion-based text generation, we evaluate **LD4LG** Lovelace et al. (2024) and **PLANNER** Zhang et al. (2023), which apply diffusion processes in the latent space of an encoder-decoder architecture.

**Datasets** We train on only the DART-MATH dataset, holding out all other benchmarks for evaluation only. Table 12 summarizes the datasets. While training is limited to mathematical reasoning problems, our evaluations also include out-of-domain tasks such as engineering and physics, providing both in-domain and out-of-domain benchmarks to assess the reasoning and generalization capabilities of LaDiR and the baselines.

### D.2 PUZZLE PLANNING

**Implementation Details** In this setting, we deliberately *disable* the answer generation and restrict the reasoning process to a single latent block, which is compressed into a fixed-size representation (4 tokens). The model is trained under a teacher-forcing regime and evaluated on decoded text tokens from our VAE, thereby isolating the latent diffusion model's capacity to capture planning dynamics without autoregressive supervision. During inference, we set the initial noise scale to 2 and the maximum diversity guidance scale to 0.8.

| Dataset | # Samples | Domain / Level | Notes |
|---|---|---|---|
| **DART-MATH** | 585k | Mixed math (train) | Synthesized for reasoning, based on GSM8K/MATH |
| **MATH** | 500 | High school / competition | In-domain benchmark |
| **GSM8K** | 1.3k | Grade school arithmetic | In-domain benchmark |
| **College-Math** | 2.8k | College-level | Linear algebra, differential equations, etc. |
| **DM-Math** | 1k | K–12 curriculum | Out-of-domain generalization |
| **OlympiaBench-Math** | 675 | Olympiad-level | Advanced competition problems |
| **TheoremQA** | 800 | STEM / theorem-driven | Math, physics, engineering |
| **Fresh-Gaokao-Math-2023** | 30 | Gaokao exam | Real-world test distribution |

Table 12: Summary of datasets used in our experiments. We use DART-MATH (Tong et al., 2024b), MATH (Hendrycks et al., 2021), GSM8K (Cobbe et al., 2021), College-Math (Tang et al., 2024b), DeepMind-Math (Saxton et al., 2019), OlympiaBench-Math (He et al., 2024), TheoremQA (Chen et al., 2023), and Fresh-Gaokao-Math-2023 (Tang et al., 2024b).

### D.3 HYPERPARAMETERS

Table 13 provides a complete summary of all hyperparameters used in our experiments, covering VAE pretraining, Stage-1 teacher-forcing training, Stage-2 rollout training, and inference settings.

| Component | Hyperparameter | Value |
|---|---|---|
| **VAE Pretraining** | Latent dimension ($d_z$) | 512 |
| | # latent tokens per block | 4 |
| | KL weight $\beta$ | $1 \times 10^{-5}$ |
| | Learning rate | $2 \times 10^{-5}$ |
| | Batch size | 128 |
| | # of Epochs | 2 |
| **Stage-1 Teacher-Forcing** | Flow-matching loss weight ($\lambda_{\mathrm{FM}}$) | 5 |
| | CE loss weight ($\lambda_{\mathrm{Ans}}$) | 1 |
| | Special-token loss weight ($\lambda_{\mathrm{Spec}}$) | 1 |
| | Learning rate | $1 \times 10^{-5}$ |
| | Batch size | 64 |
| | # of Epochs | 20 |
| **Stage-2 Rollout Training** | Learning rate | $1 \times 10^{-5}$ |
| | Batch size | 12 |
| | # of Epochs | 20 |
| **Inference** | Classifier-free guidance scale | 4 |
| | Answer token decoding temperature | 0.7 |

Table 13: Complete hyperparameter settings for all training and inference stages.

## E   ADDITIONAL MODEL DETAILS

**VAE Training Architecture**  Figure 7 is a more in-depth diagram of the training of the VAE. The encoder LLM first maps the input sequence of original text embeddings and learnable embeddings into hidden states. These hidden states are then projected through two linear layers to produce the mean and variance of the latent distribution, from which we sample the thought tokens via the reparameterization trick. The sampled latent tokens $\tilde{z}_1, \tilde{z}_2, \ldots, \tilde{z}_k$ are passed to the decoder LLM, which reconstructs the original text tokens under a teacher-forcing setup. This design enables the model to compress high-dimensional text into a smaller set of semantically meaningful latent variables, while still maintaining faithful reconstruction of the original reasoning process.

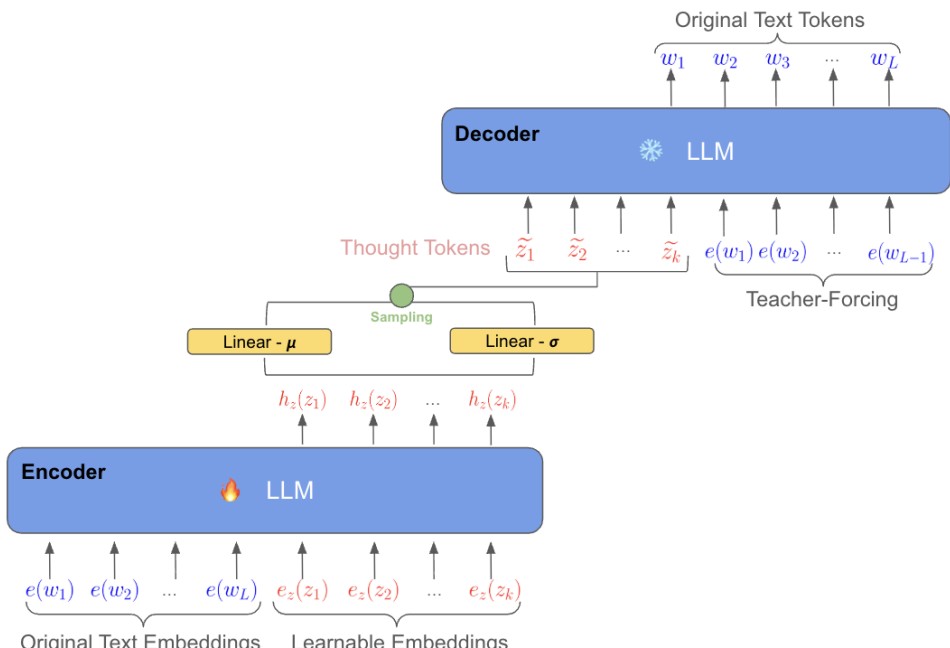

Figure 7: Detailed architecture of the variational autoencoder for latent reasoning. The encoder is a finetuned LLM that takes both original text embeddings $e(w_i)$ and learnable embeddings $e_z(z_i)$, producing mean and variance vectors through linear projections of the last hidden state $h$. Latent thought tokens $\tilde{z}_i$ are then sampled from $\mathcal{N}(\mu, \sigma^2)$. The decoder is a frozen LLM that reconstructs the original CoT text under teacher forcing, conditioned on both the sampled thought tokens and the original text embeddings.

