# OpenReview forum: "LaDiR: Latent Diffusion Enhances LLMs for Text Reasoning"
_ICLR.cc/2026/Conference — ICLR 2026 Poster_

### Official Review · Reviewer_VgbG · 2025-10-31

**Soundness:** 3
**Presentation:** 3
**Contribution:** 3
**Rating:** 6
**Confidence:** 4

**Summary:**

This paper proposes LaDiR (Latent Diffusion Reasoner), a new framework to improve the reasoning capabilities of LLMs by modeling the reasoning process in a continuous latent space using a diffusion model. The framework consists of first learning a VAE to encode the textual reasoning traces to blocks of continuous though tokens, and then a latent diffusion model to generate a sequence of blocks of latent reasoning tokens.The authors demonstrate empirically that LaDiR, using a LLaMA 3.1 8B backbone, outperforms or matches autoregressive baselines (COT and SFT) and other latent-space reasoning methods on a suite of mathematical reasoning and planning benchmarks.

**Strengths:**

1. The paper is well-motivated and situated within the growing literature on reasoning in large language models (LLMs).
2. The experiments provide convincing evidence of the method’s efficacy across multiple reasoning and planning tasks. Even though the gains over baselines are not always large or consistent, the method produces more interpretable reasoning traces (when decoded into text) and supports flexible trade-offs between inference compute and performance.
3. The technical presentation is clear and well-organized, and the figures are particularly helpful in conveying key concepts.

**Weaknesses:**

My major concern is on the computational cost and complexity of the method.
1. The multi-stage training pipeline — involving VAE pretraining followed by two stages of diffusion model training — is considerably much more involved than the baselines. Inference requires T denoising steps for B blocks, which is substantially slower than a single-pass autoregressive generation. The paper would be stronger if the paper would include a more direct comparison of the FLOPs against its baselines, and some discussion on how the induced overhead could be reduced in the future, as well as the difficulties of tuning the hyperparameters.
2. All experiments are conducted on an 8B model. It would be helpful if the model could include a more explicit discussion / analysis on how the framework scales (with data and model sizes of different components).

**Questions:**

1. Could you provide a more direct comparison of the computational cost (e.g., total FLOPs or wall-clock time) between LaDiR and the baselines?
2. The example decoded reasoning trace is surprisingly readable from Table 3 on the Countdown task. While interpretability is nice, it also seems to suggest that the latent space is still very aligned with the text space. Did you experiment with other settings that induce less interpretable latent thoughts and if so how did that impact the downstream task performance?
3. The method adopts a one sentence per block strategy. Although it seems natural for the tasks in the paper, it might not always be the optimal choice for all the cases. It could be helpful to include some analysis on how sensitive the model is to the blockization strategy.

---

> ### Author Response · Authors · 2025-11-18
> **Author Response to Reviewer VgbG [Part 1]**
>
> We appreciate your positive reviews. Our detailed responses to your comments are provided below.
>
> ## Response to Weaknesses
>
> > W1: My major concern is on the computational cost and complexity of the method. The multi-stage training pipeline — involving VAE pretraining followed by two stages of diffusion model training — is considerably much more involved than the baselines. Inference requires T denoising steps for B blocks, which is substantially slower than a single-pass autoregressive generation. The paper would be stronger if the paper would include a more direct comparison of the FLOPs against its baselines, and some discussion on how the induced overhead could be reduced in the future, as well as the difficulties of tuning the hyperparameters.
>
> **Response to W1:**
> Thank you for raising this point. We address the inference-time considerations and provide a detailed comparison in the **General Response**, where we show that LaDiR achieves competitive inference efficiency when using a moderate number of diffusion steps, and can flexibly trade off compute for accuracy.
>
> Regarding hyperparameter tuning, we note the following:
>
> - **VAE hyperparameters** (latent dimension, number of latent tokens, KL weight, $\beta$ in $\beta$-VAE etc.) are stable across tasks in our experiments, and our settings align closely with findings in prior latent diffusion work such as PLANNER. In practice, we found that these hyperparameters could generalize well without task-specific retuning.
>
> - For the **diffusion reasoning model**, the inference-time hyperparameters (e.g., classifier-free guidance scale, diversity guidance scale) behave similarly to those in diffusion models from other domains (e.g., image generation). Once the model is trained, these hyperparameters can typically be tuned once without extra training and reused across tasks.
>
> We will clarify these points and add a brief discussion on potential ways to reduce overhead in future work (e.g., fewer denoising steps, distillation, consistency models).
>
>
> > W2: All experiments are conducted on an 8B model. It would be helpful if the model could include a more explicit discussion / analysis on how the framework scales (with data and model sizes of different components).
>
>
>
> **Response to W2:**
> Thank you for the helpful suggestion. We agree that discussing scalability would strengthen the paper. While our experiments focus on an 8B model for computational feasibility, the framework is designed to scale naturally in several dimensions:
>
> - **Scaling the VAE:** The VAE encoder and decoder reuse the underlying LLM architecture, so increasing the LLM size (e.g., 8B → 14B) directly increases the expressive capacity of the latent space without changing the training pipeline.
>
> - **Scaling the diffusion reasoning model**: The diffusion model operates entirely in the latent space, whose dimension is fixed (e.g., 512). This means scaling the LLM does **not** proportionally increase the cost of diffusion training or inference. In prior latent diffusion works (e.g., PLANNER), this property has enabled stable scaling to larger LLMs.
>
> - **Scaling with data**: Both VAE pretraining and diffusion training benefit from larger datasets, and our rollout training procedure is compatible with arbitrarily large reasoning corpora. Since denoising operates over latent blocks rather than long text sequences, the cost grows slowly with dataset size.
>
> We will add a short discussion in the revision to make these scalability considerations clearer and outline future work exploring larger model scales.

---

> ### Author Response · Authors · 2025-11-18
> **Author Response to Reviewer VgbG [Part 2]**
>
> ## Response to Questions
>
> > Q1: Could you provide a more direct comparison of the computational cost (e.g., total FLOPs or wall-clock time) between LaDiR and the baselines?
>
> **Response to Q1:**
>
> Please see the inference-latency comparison in the **General Response**.
>
>
> > Q2: The example decoded reasoning trace is surprisingly readable from Table 3 on the Countdown task. While interpretability is nice, it also seems to suggest that the latent space is still very aligned with the text space. Did you experiment with other settings that induce less interpretable latent thoughts and if so how did that impact the downstream task performance?
>
> **Response to Q2:**
> Thank you for the insightful question. We would like to note that the examples shown in Table 3 use **greedy decoding**. When using **sampling with higher temperatures (e.g., 0.9)**, decoded latent thoughts naturally become *less* readable, confirming that the latent space is not constrained to fluent text. We will add a brief discussion of this observation in the revised version.
>
> > Q3: The method adopts a one sentence per block strategy. Although it seems natural for the tasks in the paper, it might not always be the optimal choice for all the cases. It could be helpful to include some analysis on how sensitive the model is to the blockization strategy.
>
> **Response to Q3:**
> Thank you for the thoughtful question. We agree that the blockization strategy is an important design choice, and we conducted preliminary experiments to study its sensitivity. We experimented with using *2 sentences per block* and *3 sentences per block*, compared to our default *1 sentence per block*.
>
> We found that increasing the number of sentences per block requires **more latent tokens** to maintain high reconstruction accuracy (around 99\%), and **increases the difficulty for the diffusion model to denoise effectively**. We did not explore sub-sentence blockization because our goal is to model *semantic reasoning steps*, not lexical fragments.
>
> Below are the results from our experiments:
>
> | # Sentences per Block | # Latent Tokens per Block | GSM8K Accuracy | MATH Accuracy |
> |-----------------------|---------------------------|----------------|---------------|
> | 1                     | 4                         | **45.2**       | **84.2**      |
> | 2                     | 8                         | 39.6           | 78.4          |
> | 3                     | 12                        | 36.1           | 72.0          |
>
> These results indicate that the model is sensitive to blockization: using too many sentences per block increases latent dimensionality and the number of latent tokens, making the diffusion process significantly harder to learn, while **one sentence per block** offers a more balanced latent size that the diffusion model can reliably denoise.

---

### Official Review · Reviewer_D4vn · 2025-11-01

**Soundness:** 1
**Presentation:** 1
**Contribution:** 1
**Rating:** 0
**Confidence:** 4

**Summary:**

This paper presents a method for text generation via latent diffusion using a block autoregressive generation strategy. They train a beta-VAE to auto-encode text. During training they incorporate a number of ad-hoc data augmentations including adding additional noise to the auto-encoder latents, and randomly replacing a token with another token randomly chosen from the vocabulary. Diffusion training requires two stages, the first is supervised training to match target latent blocks as well as generate correct answers and special token placements. Stage 2, requires rolling out latent blocks autoregressively and supervising it with the answer, backpropogating that signal through the rollout. They also propose a guidance technique for encouraging diversity among a group of samples in a batch. They benchmark against autoregressive and discrete diffusion baselines on math datasets. They also benchmark in the countdown setting which is common for evaluating planning/reasoning ability.

**Strengths:**

* The rollout training is interesting - specifically teaching the model to determine how long to think for.
* Consistent improvement over CoT in math benchmarks.
* Strong improvement over LLaMA and LLaDA on Countdown Pass@1

**Weaknesses:**

* The introduction and related work (including the related work in the appendix) carefully side steps any discussion of prior work on latent diffusion for language (yet all other topics in related work is discussed thoroughly). The current draft is incredibly misleading (almost implying they are the first to do latent language diffusion). There is a significant body of work in place that must be discussed in this paper, as this work is a direct followup to the existing work. A few of the relevant citations include,
Zhang, Y., Gu, J., Wu, Z., Zhai, S., Susskind, J., & Jaitly, N. (2023). Planner: Generating diversified paragraph via latent language diffusion model. Advances in Neural Information Processing Systems, 36, 80178-80190.
Lovelace, J., Kishore, V., Wan, C., Shekhtman, E., & Weinberger, K. Q. (2023). Latent diffusion for language generation. Advances in Neural Information Processing Systems, 36, 56998-57025.
* Further all experiments in this work only compare against autoregressive and discrete diffusion models, while the obvious and necessary comparison is the existing latent language diffusion works mentioned above. The contribution of this paper is not clear without the direct comparison to other latent language diffusion models.
* The improvements on the math benchmarks are pretty marginal, and there is no time/compute comparison provided in the paper.
* The robustness augmentations to the VAE training are ad-hoc, and they provide no ablation or study of the affects of this choice.
* This sentence "Unlike AR models that generate a single reasoning trajectory sequentially, our framework can generate multiple diverse reasoning trajectories in parallel within a batch." implies that AR models cannot do batched sampling. This is incorrect you can do batched sampling with AR models.
* The diversity improvements are not ablated. Even more concerning they are only used in the Countdown setting. If you are going to report with diversity improvements on Countdown you need to report with and without the diversity improvements in both Countdown and the mathematics benchmarks (otherwise it comes off as a trick to improve one setting).
* Pass@100 is a strange metric to report, Pass@1 is much more meaningful.
* On line 301 I believe you meant to write "... analysis in Section 4.4 provide ..."

**Questions:**

Figure 5 shows performance as a function of denoising steps for LaDiR vs. CoT, would be interesting to see wall clock times comparing the two methods. How many denoising steps can you complete in the time it takes to do CoT? Do you have a compute/time controlled comparison?

---

> ### Author Response · Authors · 2025-11-18
> **Author Response to Reviewer D4vn [Part 1]**
>
> Thank you for the review. We provide detailed clarifications and responses below.
>
> ## Response to Weaknesses
>
>
> > W1: The introduction and related work (including the related work in the appendix) carefully side steps any discussion of prior work on latent diffusion for language (yet all other topics in related work is discussed thoroughly). The current draft is incredibly misleading (almost implying they are the first to do latent language diffusion). There is a significant body of work in place that must be discussed in this paper, as this work is a direct followup to the existing work.
>
> **Response to W1:**
> Please see the  General Response to Related Work and Novelty for the detailed response for this. Also, we would like to clarify that our intention was not to imply that we are the first to explore latent language diffusion; rather, our contribution lies in adapting these ideas specifically to **latent reasoning**. In the revised version, we will explicitly cite and discuss these works, clarify their methodological connections to our framework, and clearly articulate how our adaptations extend latent diffusion to the reasoning setting.
>
>
> > W2: Further all experiments in this work only compare against autoregressive and discrete diffusion models, while the obvious and necessary comparison is the existing latent language diffusion works mentioned above. The contribution of this paper is not clear without the direct comparison to other latent language diffusion models.
>
>
> **Response to W2:** Please refer to the updated comparison with PLANNER and LD4LG on math reasoning tasks provided in the General Response to Related Work and Novelty.
>
>
>
> > W3: The improvements on the math benchmarks are pretty marginal, and there is no time/compute comparison provided in the paper.
>
> **Response to W3:**
> We would like to emphasize that our method achieves both higher accuracy (Pass@1) and greater reasoning diversity (Pass@100). As shown in the updated results in the General Response to Common Weakness 2, our approach yields a **6.03%** absolute gain in pass@100 over the AR SFT baseline, demonstrating a clear advantage. Updated inference-latency measurements are also provided in the General Response to Common Weakness 1.
>
>
> > W4: The robustness augmentations to the VAE training are ad-hoc, and they provide no ablation or study of the affects of this choice.
>
> **Response to W4:**  We vary the standard deviation $k$ of latent Gaussian noise and the token substitution probability $p$. Best performance is achieved at $k = 3$ and $p = 0.3$.
>
> ### Latent Gaussian Noise Ablation with p=0.3
>
> | k (std of noise) | GSM8K Acc (%) |
> |------------------|----------------|
> | 0                |     68.3       |
> | 1                |     73.4       |
> | 3                |     84.2       |
> | 5                |     79.4       |
>
> ### Input Token Substitution Ablation with k = 3
>
> | p (substitution prob.)  | GSM8K Acc (%)  |
> |-------------------------|----------------|
> | 0.0                     |      70.2      |
> | 0.1                     |      78.3      |
> | 0.3                     |      84.2      |
> | 0.5                     |      64.0      |
> | 0.7                     |      32.4      |
>
> > W5: This sentence "Unlike AR models that generate a single reasoning trajectory sequentially, our framework can generate multiple diverse reasoning trajectories in parallel within a batch." implies that AR models cannot do batched sampling. This is incorrect you can do batched sampling with AR models.
>
> **Response to W5:** Please see the General Response to this comment. Thank you.
>
> > W6: The diversity improvements are not ablated. Even more concerning they are only used in the Countdown setting. If you are going to report with diversity improvements on Countdown you need to report with and without the diversity improvements in both Countdown and the mathematics benchmarks (otherwise it comes off as a trick to improve one setting).
>
> **Response to W6:** Please see the General Response for the updated results of diversity improvement on math reasoning tasks.
>
> > W7: Pass@100 is a strange metric to report, Pass@1 is much more meaningful.
>
> **Response to W7:** We would like to clarify that we report both **Pass@1** and **Pass@100** in the paper (Table 2). Also, Pass@100 provides a complementary view of the model’s **diversity and exploration ability**. As noted in Lines 402–403, high Pass@k highlights the model’s potential for **reinforcement learning–style post-training**, where diverse candidate trajectories are valuable training signals.
>
> > W8: On line 301 I believe you meant to write "... analysis in Section 4.4 provide ..."
>
> **Response to W8:** Thank you for catching this. We will add the missing “in” in the revised version.

---

> ### Author Response · Authors · 2025-11-18
> **Author Response to Reviewer D4vn [Part 2]**
>
> ## Response to Questions
>
> > Q1: Figure 5 shows performance as a function of denoising steps for LaDiR vs. CoT, would be interesting to see wall clock times comparing the two methods. How many denoising steps can you complete in the time it takes to do CoT? Do you have a compute/time controlled comparison?
>
> **Response to Q1:** Please see the General Response for the updated results on inference latency.

---

### Official Review · Reviewer_nMv9 · 2025-11-01

**Soundness:** 1
**Presentation:** 1
**Contribution:** 2
**Rating:** 0
**Confidence:** 5

**Summary:**

This work adapts latent language diffusion to reasoning tasks. This work follows the typical pattern of first training a language autoencoder and then training a diffusion model in that continuous space. This work then trains a reasoning model which sequentially diffuses latent representations of textual thoughts before producing the final answer. They evaluate their approach for mathematical reasoning tasks and a synthetic puzzle task.

**Strengths:**

This work focuses on an interesting area (latent reasoning) that is gaining interest in the prior year.

One big problem with prior latent reasoning approaches (e.g. Coconut) is that it is challenging to appropriately supervise the latents. Diffusion modeling provides an elegant solution to that problem.

The authors report good results across both math reasoning tasks and a synthetic puzzle task. The visualization of the decoding process provides some nice intuition about the decoding process.

**Weaknesses:**

This paper cannot be accepted without properly situating itself within the literature. This work extensively discusses discrete diffusion models and recent latent reasoning work, but completely omits any discussion of the most methodologically relevant work: continuous diffusion models for language generation. Although this work falls short of claiming to introduce continuous/latent language diffusion, they also fail to acknowledge or discuss prior work that has explored this approach. A reader unfamiliar with the area would reasonably be given the impression that this approach is entirely unexplored before this current submission.

Works like Diffusion-LM, LD4LG, PLANNER, have already explored continuous diffusion for language generation across a variety of tasks, the latter two exploring latent diffusion in a very similar setup to this work. These are not obscure papers. They are highly cited papers published at top conferences 1+ years ago. Basic searches (e.g. “latent language diffusion”) return these papers as top results. In particular, the autoencoder design in the current submission seems to follow PLANNER very closely. The core technical approach (i.e. autoencoder-based latent diffusion for text) is established in prior work. This paper’s contribution is applying this approach to reasoning tasks, which is potentially valuable but must be communicated clearly.

This is not an isolated oversight. The related works section is otherwise quite comprehensive and discusses discrete diffusion and latent reasoning at length, but completely omits any mention of continuous/latent language diffusion. This gap covers the most methodologically relevant prior work.

There are various downstream weaknesses that stem from this omission. Given that this is primarily an application of an existing approach, the framing in the intro and in Figure 1 gives an uninformed reader an incorrect impression of the contribution of this work.
An unnecessary amount of space is given to motivating and describing the VAE design when it is almost the same as the PLANNER VAE architecture.

There are other weaknesses that can be discussed, but the primary issue with this work is the above mentioned. The combination of a complete omission of the most methodologically relevant prior work and a framing that implies novelty for an existing approach warrants rejection.

The description of the diffusion and autoregressive architecture and how precisely they are trained is unclear. The figure makes them appear to be one model, but the text uses f() and p() to denote the diffusion and autoregressive models, respectively, with different parameters. If they are one model, the training procedure is unclear given that the autoregressive model and diffusion model receive different inputs (clean vs noisy latent thoughts). Are they simply multi-tasked? Are there two separate models as the notation, but not the figure, implies?

This statement is untrue:
“Unlike AR models that generate a single reasoning trajectory sequentially, our framework can generate multiple diverse reasoning trajectories in parallel within a batch“
Autoregressive models are also capable of batched inference.

The modification to improve diversity seem somewhat arbitrary and are only used to drive up the PASS@100 metric for the Countdown task which of course benefits significantly from increasing diversity. The benefits of these changes are not demonstrated elsewhere.

Many training and implementation details are omitted. How long were various models trained for? With what hyperparameter settings, etc?

Discussion of inference latency is not present. Inference should be much slower than the purely autoregressive baselines.

**Questions:**

I do not have additional questions.

---

> ### Author Response · Authors · 2025-11-18
> **Author Response to Reviewer nMv9**
>
> Thanks for your time and our detailed responses to your comments are provided below.
>
> ## Response to Weaknesses
>
> > W1: The combination of a complete omission of the most methodologically relevant prior work and a framing that implies novelty for an existing approach warrants rejection.
>
> **Response to W1:** Please see the General Response on Related Work and Novelty .
>
>
> > W2: The description of the diffusion and autoregressive architecture and how precisely they are trained is unclear. The figure makes them appear to be one model, but the text uses f() and p() to denote the diffusion and autoregressive models, respectively, with different parameters. If they are one model, the training procedure is unclear given that the autoregressive model and diffusion model receive different inputs (clean vs noisy latent thoughts). Are they simply multi-tasked? Are there two separate models as the notation, but not the figure, implies?
>
> **Response to W2:**
> Thanks for your feedback. To clarify concisely: **there is only one model**, not two. The diffusion component and the autoregressive component are simply **two heads on the same LLM backbone**, sharing all transformer layers and parameters $\psi$. The figure shows a unified model because the backbone is shared; the notation $f_\psi(\cdot)$ and $p_\psi(\cdot)$ was meant only to distinguish the two *functions* the same model performs, not separate networks.
>
> The training is **not multi-tasking two independent models**. Instead:
>
> - During diffusion training, the model receives **noisy latent blocks** and the flow-matching head predicts denoised latent states.
> - During answer-token and special-token prediction, the **same backbone** receives **clean/denoised latent blocks** (teacher-forced in Stage 1; self-generated in Stage 2) and the language-modeling heads predict next discrete tokens.
>
> We will revise the texts and notations to make this more clear in the next version.
>
> > W3: This statement is untrue: “Unlike AR models that generate a single reasoning trajectory sequentially, our framework can generate multiple diverse reasoning trajectories in parallel within a batch“ Autoregressive models are also capable of batched inference.
>
> **Response to W3:** Please see the General Response.
>
> > W4: The modification to improve diversity seem somewhat arbitrary and are only used to drive up the PASS@100 metric for the Countdown task which of course benefits significantly from increasing diversity. The benefits of these changes are not demonstrated elsewhere.
>
> **Response to W4:**
> Thank you for the comment. We would like to clarify that our diversity-improvement mechanism is **not** tailored specifically for Countdown; as shown in the General Response, our updated results demonstrate clear diversity gains on **math reasoning tasks** as well.
>
> As noted in Lines 402–403, **high Pass@k indicates strong potential for reinforcement learning–style post-training**, where diverse trajectories enable sampling better training examples in rollout. We view this as an important future direction and will clarify the general applicability of the diversity mechanism in the revised version.
>
> > W5: Many training and implementation details are omitted. How long were various models trained for? With what hyperparameter settings, etc?
>
> **Response to W5:**
>
> Thank you for pointing this out. We will expand the training and implementation details in the revised version. Specifically, we will add a full table summarizing all key hyperparameters and training configurations for (1) VAE pretraining, (2) Stage-1 teacher-forcing training, and (3) Stage-2 rollout training, as well as inference-time settings. Below is the table we will include:
>
> | Component | Hyperparameter | Value |
> |----------|----------------|-------|
> | **VAE Pretraining** | Latent dimension ($d_z$) | 512 |
> |  | # latent tokens per block | 4 |
> |  | KL weight $\beta$ | 1e-5 |
> |  | Learning rate | 2e-5 |
> |  | Batch size | 128 |
> |  | # of Epochs | 2 |
> | **Stage-1 Teacher-Forcing** | Flow-matching loss weight ($\lambda_{\mathrm{FM}}$) | 5 |
> |  | CE loss weight ($\lambda_{\mathrm{Ans}}$) | 1 |
> |  | Special-token loss weight ($\lambda_{\mathrm{Spec}}$) | 1 |
> |  | Learning rate | 1e-5 |
> |  | Batch size | 64 |
> |  | # of Epochs | 20 |
> | **Stage-2 Rollout Training** | Learning rate | 1e-5 |
> |  | Batch size | 12 |
> |  | # of Epochs | 20 |
> | **Inference** |  CFG scale | 4 |
> |  | Decoding strategy for answer tokens | Temperature=0.7 |
>
> We will include this detailed table in the revised paper to make all training settings transparent and reproducible.
>
>
> > W6: Discussion of inference latency is not present. Inference should be much slower than the purely autoregressive baselines.
>
> **Response to W6:** Please see the General Response for the updated results on inference latency.

---

### Official Review · Reviewer_FWxU · 2025-11-01

**Soundness:** 3
**Presentation:** 3
**Contribution:** 3
**Rating:** 6
**Confidence:** 3

**Summary:**

This paper studies LaDiR (Latent diffusion for reasoning), where the model utilizes blocked latent thoughts generated by diffusion methods and generates the final answer conditioned on these latent thoughts. The latent thoughts used for training are generated by training a VAE (variational autoencoder), which encodes the original textual CoT to latent representations. Comparisons on numerous benchmarks with several baseline methods (including SFT+CoT and several latent reasoning methods) prove the advantages of the proposed methods.

**Strengths:**

1. Using diffusion methods to generate latent thoughts is a good idea.
2. The proposed method can flexibly control both the computational budget and the diversity of trajectories.
3. The decoder of VAE makes the latent thought interpretable.

**Weaknesses:**

1. For many equations, it would be better if the authors could define the function and variables more rigorously and clearly (e.g., by providing the shape of inputs or outputs, as discussed in the question section).
2. In lines 35-38, the authors argued that “AR models generate a single linear chain of thought (CoT), which limits reasoning diversity and restricts exploration of multiple valid solutions”. However, AR models can also generate continuous CoT, which can explore multiple solutions in superposition as theoretically demonstrated by [1]. I think it’s worth discussing it or changing the wording to, e.g., “AR models with discrete CoT are limited by …”.
3. In line 95, equation (1), I think the notation and definition of each function or variable should be made clearer and more rigorous. For example, what is $p\_{\theta}$, $p(\cdot)$, and what’s the shape of their input and output? What is the shape of x and z, and what’s the distribution of $x$?

**References**:

[1] Zhu, Hanlin, Shibo Hao, Zhiting Hu, Jiantao Jiao, Stuart Russell, and Yuandong Tian. "Reasoning by Superposition: A Theoretical Perspective on Chain of Continuous Thought." arXiv preprint arXiv:2505.12514 (2025).

**Questions:**

1. When you are generating the next block, will you remove the timestamp from the previous block?
2. I’m a bit confused by Equation (4). According to your notation, $Z^{(<=B)}$ should be the whole thought, and $y\_{\leq \tau}$ is a prefix of the final answer. Why will $y_{\tau}$ be a $< EOT >$ token since all $< EOT >$ should appear in $Z^{(<=B)}$. Am I missing anything?
3. In line 294, what is the definition of $x\_t$ and $x$, and what are their shape?
4. In line 342, should it be 1\% instead of 2\%? (41.8 vs 40.8)
5. In line 344, it seems that for DM-Math, CoT outperforms LaDiR.

---

> ### Author Response · Authors · 2025-11-18
> **Author Response to Reviewer FWxU [Part 1]**
>
> Thank you for your positive feedback. We have provided detailed responses to your comments below.
>
> ## Responses to Weaknesses
>
> > W1: For many equations, it would be better if the authors could define the function and variables more rigorously and clearly (e.g., by providing the shape of inputs or outputs, as discussed in the question section).
>
> **Response to W1:**
> Thanks for your feedback, and we will make sure to define the function and variables more rigorously and clearly. Please see the detailed explanations in the answers to your questions.
>
>
> > W2: In lines 35-38, the authors argued that “AR models generate a single linear chain of thought (CoT), which limits reasoning diversity and restricts exploration of multiple valid solutions”. However, AR models can also generate continuous CoT, which can explore multiple solutions in superposition as theoretically demonstrated by [1]. I think it’s worth discussing it or changing the wording to, e.g., “AR models with discrete CoT are limited by …”.
>
> **Response to W2:**
> Thank you for the suggestion. We would like to clarify that we  specify discrete CoT in Line 35–38. While Coconut-like method enable AR models to theoretically represent continuous CoT in superposition [1], they use LLM's hidden states as latent tokens which are **deterministic** and not sampled (see the Pass@100 results in the General Response to Common Weakness 2). In contrast, our diffusion-based latent reasoning introduces **stochastic latent tokens explicitly sampled from a distribution** and denoised through diffusion steps, enabling *exploration of multiple reasoning trajectories*. This distinction between deterministic hidden states (Coconut) and sampled latent reasoning distributions (ours) underlies our claim about reasoning diversity. We will make this more clear in the revision.
>
> > W3: In line 95, equation (1), I think the notation and definition of each function or variable should be made clearer and more rigorous. For example, what is $p_\theta$, $p(\cdot)$, and what’s the shape of their input and output? What is the shape of x and z, and what’s the distribution of $x$?
>
> **Response to W3:**
> Thank you for the suggestion. We have provided detailed definitions in **Appendix Section B.1**, and we will further revise the notation to make it clearer in the main text. To answer your question, the definitions are as follows:
>
> - $x \in \mathbb{R}^{L \times d_x}$: a sequence of text token embeddings with sequence length $L$ and embedding dimension $d_x$ (same as the LLM hidden size, e.g., 4096 for LLaMA 3.1 8B).
> - $z \in \mathbb{R}^{M \times d_z}$: latent representations produced by the encoder, where $M$ is the predefined number of latent tokens and $d_z$ is the latent dimension (512 in our implementation).
> - $p_\theta(x|z)$: decoder likelihood parameterized by $\theta$, reconstructing $x$ from the latent variables $z$.
> - $q_\phi(z|x)$: approximate posterior distribution defined by the encoder, parameterized by $\phi$.
> - $p(z)$: prior distribution over latents, typically a standard Gaussian $\mathcal{N}(0, I)$.
> - $x \sim p_{\text{data}}(x)$: data samples drawn from the training corpus (e.g., a single reasoning step in our case).
>
> We will ensure these notations and shapes are explicitly clarified in the revised version for completeness.

---

> ### Author Response · Authors · 2025-11-18
> **Author Response to Reviewer FWxU [Part 2]**
>
> ## Response to Questions
> >Q1: When you are generating the next block, will you remove the timestamp from the previous block?
>
> **Response to Q1:**
> Yes. As illustrated in **Figure 2**, the timestamp from the first block is **removed** when generating the second block. Each new block starts with a clean/denoised context to ensure the diffusion process models the reasoning progression independently of prior timestamps.
>
> > Q2: I’m a bit confused by Equation (4). According to your notation, $\mathbf{Z}^{(\le B)}$ should be the whole thought, and $y_{\le \tau}$ is a prefix of the final answer. Why will $y_{\le \tau}$ be a <EOT\> token since all <EOT\> should appear in $\mathbf{Z}^{(\le B)}$. Am I missing anything?
>
> **Response to Q2:**
> Thank you for the thoughtful question. To clarify, the actual special token loss does **not** condition on any $y_{\le \tau}$ terms. Its formulation should be as:  $$
> L_{\text{spec}}
> = - \sum_{\tau \in T_{\text{eot}}} \log p(s_{\tau} \mid q, Z_{\le \tau}).
> $$ This objective supervises the model to correctly predict **special tokens** (e.g., <BOT>) given the query $q$ and the latent reasoning blocks before token position $\tau$. In essence, it helps the model learn *when* to terminate reasoning based on the latent context $\mathbf{Z}^{(\le \tau)}$. We will further clarify this explanation and notation in the revised version for better readability.
>
> >Q3: In line 294, what is the definition of $x$ and $x_t$, and what are their shape?
>
> **Response to Q3:**
> Thank you for the question. In this context, $x$ denotes the **conditioning information** used during diffusion—this includes the question tokens and all preceding latent reasoning blocks—while $x_t$ represents the **noisy latent state** at diffusion step $t$.
>
> Their shapes are as follows:
>
> - **Noisy latent state:** $x_t \in \mathbb{R}^{M \times d_z}$
>   - $M$: number of latent tokens
>   - $d_z$: latent dimension (512 in our implementation)
>
> - **Context**: $x \in \mathbb{R}^{K \times d_x}$
>   - $K$: total number of contextual tokens (question tokens + preceding projected latent blocks)
>   - $d_x$: LLM hidden size (e.g., 4096 for LLaMA 3.1 8B)
>   - All contextual tokens are projected into this shared dimension before being passed into the diffusion model.
>
> To improve clarity, we will rename $x$ to **$c$** (for “context”) in the revised version and update the notation accordingly in the next revision.
>
> > Q4:In line 342, should it be 1% instead of 2%? (41.8 vs 40.8)
>
> **Response to Q4:**
> Thank you for pointing this out. The reported value used **relative percentage improvement**, which is calculated as: (41.8 - 40.8)/40.8 = 2.4%
>
> However, we agree that using **absolute percentage improvement** (i.e., 41.8 − 40.8 = **1.0%**) would make the comparison clearer for readers. We will update the text to explicitly state the metric being used and revise the phrasing to avoid confusion.
>
> > Q5: In line 344, it seems that for DM-Math, CoT outperforms LaDiR.
>
> **Response to Q5:**
> Thank you for the careful reading. You are correct, and the sentence in line 344 was intended to refer to the **Gaokao** dataset, where LaDiR shows improvement over CoT. We will revise the wording and avoid this confusion in the updated version.

---

### Author Response · Authors · 2025-11-18
**Response to Reviewer D4vn and nMv9's Concerns on Related Work and Novelty [Part 2]**

### 3. Empirical Comparisons and Planned Revisions

To provide a direct and controlled comparison with existing latent diffusion approaches, we re-ran **LD4LG** and **PLANNER** under the same experimental setting. Specifically, we used the same **LLaMA-3.1-8B** model for the VAE decoder for all methods, **FLAN-T5** as the encoder for LD4LG and PLANNER, and trained all methods on the same reasoning datasets. For LD4LG and PLANNER, following the implementation in the original papers, we encoded the **answer paragraph** into a single latent representation (i.e., no block diffusion).

The results are summarized below:

| Method        | MATH | GSM8K |
|---------------|------|--------|
| **PLANNER**   | 5.7  | 18.7   |
| **LD4LG**     | 9.1  | 32.9   |
| **LaDiR (ours)** | **45.2** | **84.2** |

These results highlight a big performance gap: while LD4LG and PLANNER prioritize **fluency** of text generation, they do not effectively support **multi-step reasoning**. In contrast, LaDiR is explicitly designed for latent **reasoning**, resulting in significantly stronger performance on reasoning benchmarks.

To address the feedback comprehensively:

- We will add a new paragraph in Related Work comparing our approach with Diffusion-LM, LD4LG, and PLANNER (with citations and distinctions summarized as above).

- We will reframe the Introduction and Figure 1 to emphasize our focus on latent reasoning rather than architectural novelty.

- We will add an additional baseline comparison to the aforementioned latent diffusion text generation methods in the main paper, acknowledging the methodological similarities.

We hope these clarifications and additions will substantially strengthen the paper’s positioning and transparency.

---

### Author Response · Authors · 2025-11-18
**Response to Reviewer D4vn and nMv9's Concerns on Related Work and Novelty [Part 1]**

We thank the reviewers for bringing up the prior works on continuous/latent diffusion models for language generation (e.g., Diffusion-LM [Li et al., ICLR 2022], LD4LG [Lovelace et al., NeurIPS 2023], PLANNER [Zhang et al., NeurIPS 2023]). We recognize that these papers are related to our work and we will revise the Related Work and Introduction sections to explicitly include these works. Also, we appreciate the opportunity to clarify both the relationship and difference between our approach and these prior methods.

### 1. Clarifying Relationship to Continuous/Latent Diffusion for Text Generation
 In particular:

- Diffusion-LM first introduced continuous diffusion for text generation in the embedding space;
- LD4LG and PLANNER proposed autoencoder-based latent diffusion for text, where a VAE first maps text into a latent space and a diffusion model diffuse/denoise in this space, which finally *requires a VAE decoder to reconstruct fluent text outputs* after denoising;
- Our model builds on this general paradigm but **targets a fundamentally different goal: reasoning rather than generation**.

Unlike prior works, which aim to improve text quality or diversity, our focus is on latent reasoning — modeling the reasoning process itself as a continuous diffusion over latent thought representations, rather than over token or embedding spaces. More importantly, **we emphasize the ability of latent tokens that lead to the correct answer**, as captured through our rollout training procedure: we explicitly propagate training signals from answer tokens back to the latent tokens that generated them. **This setup allows the diffusion process to learn which latent thoughts contribute to correct reasoning outcomes, rather than merely producing fluent text.** This reframing changes both the training objective — from text likelihood to reason consistency and answer alignment — and the evaluation protocol — from fluency to reasoning correctness on multi-step reasoning tasks.

### 2. Novelty of Our Work

While the underlying autoencoder–diffusion structure is similar in spirit to PLANNER and LD4LG, adapting latent diffusion to reasoning imposes several important changes. Prior latent diffusion works are designed for *text generation*, where the objective is to reconstruct or generate fluent text. In contrast, our focus is **latent reasoning**, which *requires learning latent trajectories that causally lead to correct answers, not textual fluency*. To make latent diffusion suitable for reasoning tasks, we introduce the following key adaptations:

- **Variable-length continuous latent diffusion via blockwise reasoning:** Prior latent diffusion works such as PLANNER and LD4LG generate an entire sentence or paragraph from a single *fixed-sized* latent representation. In contrast, we model each reasoning step as a separate block and introduce a semi-autoregressive blockwise diffusion process that allows the model to *learn causal dependencies between reasoning steps*. This design naturally supports *variable-length* Chain-of-Thought (CoT) reasoning for tasks of different complexity. Moreover, unlike discrete block diffusion, continuous block diffusion suffers more from *error accumulation across steps*; *we address this challenge through a 2nd-stage rollout training* procedure that explicitly mitigates accumulated errors and stabilizes long-horizon reasoning.

- **Improved efficiency and reduced error by avoiding unnecessary decoding:** Prior latent-diffusion text generation methods (e.g., PLANNER, LD4LG) must decode each intermediate latent back into text to evaluate generation quality or guide the next step. This repeated latent→text→latent conversion introduces **rounding and discretization errors**, which accumulate over long reasoning chains and can degrade performance. In contrast, our latent space is explicitly designed for reasoning. We only decode latent thoughts into text when interpretability is needed, rather than at every diffusion step. **This not only reduces computational cost by avoiding repeated decoder forward passes, but also prevents the accumulation of rounding error from repeatedly mapping between continuous latents and discrete tokens.** Conceptually, this aligns with recent large reasoning models (e.g., GPT-5), which keep most internal reasoning hidden—but our approach achieves this with even greater computational efficiency.

We will make this framing explicit to emphasize the novelty and architectural design tailored for reasoning tasks. The revision will clarify that our work represents an application and extension of latent diffusion modeling to the reasoning domain, introducing new learning objectives and empirical insights specific to this setting.

---

### Author Response · Authors · 2025-11-18
**General Response to Common Weaknesses**

We thank all reviewers for their time and feedback. We would like to address several common weaknesses from reviewers below:

### Common Weakness 1 (from Reviewer nMv9, D4vn and VgbG):

> Discussion of inference latency is not present. Inference should be much slower than the purely autoregressive baselines.

**Response to Common Weakness 1:**
Thank you for highlighting the missing latency comparison. We have now added **inference latency results** on the MATH dataset using the same evaluation batch size (8) and identical hardware:

| Method | Avg Wall-Clock Time per Example | Pass@1 | Pass@100 |
|--------|--------------------------------|--------|----------|
| LLaMA CoT SFT | 4.7 s | 43.1 | 89.0 |
| **LaDiR (10 steps)** | 4.9 s | 42.9 | 90.7 |
| **LaDiR (30 steps)** | 14.8 s | 44.9 | 92.8 |

These results show that LaDiR can **match the latency** of autoregressive baselines at **10 diffusion steps**, while enabling a flexible compute–performance trade-off at more inference steps. This efficiency stems from the fact that **each latent block contains only 4 latent tokens**, representing on average **~22 text tokens**. Such compression reduces per-step computation and shortens the preceding context for later blocks—resulting in a lower cumulative cost than autoregressive decoding, which repeatedly processes long text-token contexts.


We will include these updated inference latency results in the revised version.

### Common Weakness 2 (from Reviewer nMv9 and D4vn):
> Diversity improvement in math reasoning tasks is unclear.

**Response to Common Weakness 2:**
Thank you for raising this concern. Please see the updated Pass@100 results for math reasoning tasks alongside Pass@1 in a single combined table below.

| Method | MATH (P@1 / P@100) | GSM8K (P@1 / P@100) | Gaokao (P@1 / P@100) | DM-Math (P@1 / P@100) | College (P@1 / P@100) | Olympia (P@1 / P@100) | TheoremQA (P@1 / P@100) | Avg (P@1 / P@100) |
|--------|----------------------|----------------------|------------------------|-------------------------|-------------------------|--------------------------|---------------------------|--------------------------|
| LLaDA CoT SFT | 39.0 / 59.9 | 82.3 / 92.4 | 20.1 / 43.0 | 43.7 / 50.7 | 38.9 / 58.3 | 5.9 / 10.2 | 20.9 / 26.3 | 35.83 / 48.69 |
| LLaMA CoT SFT | 43.1 / 49.8 | 84.5 / 89.0 | 30.7 / 37.9 | 47.8 / 52.0 | 45.7 / 54.9 | 10.1 / 12.9 | 21.2 / 25.0 | 40.44 / 45.93 |
| Coconut | 37.3 / 39.3 | 68.3 / 74.3 | 26.8 / 29.3 | 33.5 / 36.9 | 40.2 / 42.9 | 5.8 / 6.3 | 11.4 / 14.9 | 31.90 / 34.84 |
| Discrete Latent | 43.2 / 47.3 | 83.9 / 88.6 | 33.3 / 39.7 | 44.7 / 49.5 | 47.1 / 53.7 | 13.3 / 17.8 | 20.3 / 28.5 | 40.83 / 46.44 |
| **LaDiR** | **45.2 / 63.7** | **84.2 / 93.7** | **33.4 / 45.8** | **46.3 / 54.2** | **48.6 / 60.3** | **11.9 / 15.3** | **22.9 / 30.7** | **41.79 / 51.96** |

**Key takeaways:**
1. **LaDiR achieves the highest Pass@100 across math datasets**, showing that its diversity-improving mechanism **generalizes beyond Countdown** and consistently produces more diverse reasoning trajectories.
2. **Coconut’s deterministic hidden-state latents** limit its ability to generate diverse solutions, which is reflected in its marginal increase from Pass@1 to Pass@100.
3. **Standard AR SFT baselines (i.g., LLaMA CoT SFT)** also show relatively low Pass@100, indicating that *batched inference alone does not lead to better diversity in reasoning*.


We will consolidate these results in the revision and clarify both the accuracy and diversity benefits of LaDiR.

### Common Weakness 3 (from Reviewer nMv9 and D4vn):
> "Unlike AR models that generate a single reasoning trajectory sequentially, our framework can generate multiple diverse reasoning trajectories in parallel within a batch." Autoregressive models are also capable of batched inference."


**Response to Common Weakness 3:**
We would like to clarify that this is a **misunderstanding**: although AR models can perform *batched inference*, batching alone does **not** guarantee **diverse** trajectories. In practice, AR models often produce highly similar reasoning traces across the batch—even at higher sampling temperatures (0.8), as shown for LLaMA 8B in Table 2 and the Response to Common Weakness 2 above.

In contrast, LaDiR includes an explicit **diversity-guidance mechanism** during inference (Lines 279–294), where a **repulsion term** encourages each latent trajectory to move away from others in the batch. This ensures that parallel trajectories explore *distinct* regions of the latent space rather than collapsing to similar solutions.

---

### Author Response · Authors · 2025-11-27
**Follow-up to all Reviewers**

Dear Reviewers,

I hope this message finds you well. As the discussion period is nearing its end with less than 6 days remaining, I would like to ensure we have addressed all your concerns satisfactorily. If there are any additional points or feedback, please feel free to let us know. Your insights are invaluable to us, and we are eager to address any remaining issues to improve our work. Thank you for your time and effort in reviewing our paper.

---

### Author Response · Authors · 2025-12-03
**Final Remarks from Authors**

We sincerely thank all reviewers for their thoughtful and constructive feedback. We are encouraged that the reviewers recognized the novelty and potential of our framework.

### Common Strengths Highlighted by Reviewers
Several common strengths were summarized across the reviews:

* **1. Timely Focus on Latent Reasoning with Diffusion:** Reviewers found the focus on **latent reasoning** timely and impactful. They highlighted that using *diffusion to generate and supervise latent thoughts* provides an elegant solution to the supervision challenges faced by prior approaches such as Coconut (nMv9, VgbG).
* **2. Interpretability and Flexible Control:** The *VAE decoder enables interpretable latent thoughts* (FWxU, VgbG, nMv9), and the diffusion framework allows *flexible control over both computational budget and trajectory diversity* (FWxU, VgbG). Reviewers also noted the value of *rollout training*, which effectively teaches the model *how long to think* (D4vn).
* **3. Strong Empirical Results and Presentation:** Reviewers emphasized the *consistent improvements over CoT*, the *strong gains on Countdown*, and the *convincing results across math and planning tasks* (D4vn, VgbG). They also appreciated the *clear technical presentation* and *helpful figures* that clarify the method (VgbG).

### Summary of Key Revisions
We have addressed all concerns thoroughly in the revised manuscript. Key updates include:

* **Comparison with Prior Latent Diffusion Works:** Addressing concerns from *D4vn (W1, W2)* and *nMv9 (W1)* regarding prior work, we added explicit comparisons to **LD4LG** and **PLANNER**. Experiments in *Section 4.1 (Table 2)* demonstrate that LaDiR significantly outperforms these baselines (e.g., **45.2% vs 5.7%** on MATH), confirming that our architecture targets **reasoning correctness** rather than just textual fluency. We also expanded **Section 5 (Related Works)** to discuss latent diffusion in text generation.
* **Inference Latency Analysis:** Addressing concerns from **nMv9 (W6)**, *D4vn (W3, Q1)*, and *VgbG (W1)* regarding computational cost, we added a latency comparison table in **Section 4.6**. Results show that LaDiR matches the speed of autoregressive baselines at lower diffusion steps (e.g., **4.9s vs 4.7s**) while offering a flexible trade-off between compute and accuracy at higher steps.
* **Expanded Diversity Evaluation:** Addressing concerns from *nMv9 (W4)* and *D4vn (W6)* regarding the generalizability of diversity gains, we extended our diversity analysis beyond the Countdown task. New **Pass@100** results across all math benchmarks in **Section 4.1 (Table 2)** show that LaDiR consistently achieves the highest diversity scores, demonstrating that our *repulsion guidance mechanism* generalizes well.
* **Notation Polishment:** Addressing *FWxU’s (W1, W3)* request for rigor, we rigorously polished the *mathematical notation* in **Section 3** and **Appendix B.1**, explicitly defining variable shapes and distinguishing between context and latent states to improve clarity.
* **Extended Ablation Studies:** Addressing *D4vn’s (W4)* and *VgbG’s (Q3)* comments on design choices, we incorporated comprehensive ablation studies in **Section 4.4** and **Appendix D**. This includes analysis on *VAE robustness augmentations* (noise scale and token substitution), the impact of *Stage-2 rollout training*, and the sensitivity of the *blockization strategy*.
* **Enhanced Reproducibility and Clarity:** Addressing *nMv9’s (W2, W5)* and *VgbG’s (W1)* feedback on implementation details, we clarified the unified single-model architecture in **Section 3** and added a comprehensive table of hyperparameters in **Appendix D** covering VAE pretraining, Teacher-Forcing, and Rollout training stages to ensure full reproducibility.

---

### Meta-Review · Area_Chair_Np9g · 2026-01-07

**Summary:**

This paper proposes LaDiR, a latent-diffusion reasoning framework that encodes textual CoT into blockwise latent "thoughts" via a VAE and then uses diffusion sampling in latent space to generate diverse latent trajectories before decoding an answer. Reviewers agree the direction is timely and the empirical results are promising, with interpretability via the VAE decoder and controllable compute/diversity as clear strengths. However, the submission initially had serious issues: weak positioning relative to continuous/latent language diffusion prior work, unclear architectural/training descriptions, and missing cost/latency accounting -- two reviewers even recommended strong reject on these grounds. The rebuttal substantially improves the paper by adding explicit comparisons to PLANNER/LD4LG, a latency table, expanded diversity evaluation beyond a single task, cleaned-up notation, and additional ablations.

**Reviewer Concerns:**

Concerns adequately addressed -

* Added discussion and direct comparisons to PLANNER and LD4LG, and expanded related works accordingly.
* Added an inference latency comparison with diffusion-step trade-offs; clarified efficiency regimes where LaDiR is comparable to AR baselines.
* Extended diversity analysis beyond Countdown and reported Pass@K across math benchmarks.

Remaining concerns -

* Multi-stage pipeline (VAE + diffusion training stages + rollout) and diffusion inference steps introduce tuning burden and potential brittleness versus simpler AR/RL baselines.
* Most results focus on an 8B backbone; scaling behavior with larger models/datasets remains underexplored relative to claimed generality.

**Reviewer Scores:**

I dont think so.

---

### Decision · Program_Chairs · 2026-01-26

Accept (Poster)